# Quantifying the impact of uncertainty on threat management for biodiversity

Sam Nicol [1], James Brazill-Boast[2], Emma Gorrod[2,3], Adam McSorley[2], Nathalie Peyrard[4] & Iadine Chadès [1]

With inadequate resources to manage the threats facing biodiversity worldwide, achieving projected management outcomes is critical for efficient resource allocation and species recovery. Despite this, conservation plans to mitigate threats rarely articulate the likelihood of management success. Here we develop a general value of information approach to quantify the impact of uncertainty on 20 threatening processes affecting 976 listed species and communities. To our knowledge, this is the most comprehensive quantification of the impacts of uncertainty on threat management. We discover that, on average, removing uncertainty about management effectiveness could triple the gain in persistence achieved by managing under current uncertainty. Management of fire, invasive animals and a plant pathogen are most impeded by uncertainty; management of invasive plants is least impacted. Our results emphasise the tremendous importance of reducing uncertainty about species responses to management, and show that failure to consider management effectiveness wastes resources and impedes species recovery.

[1] CSIRO, Ecosciences Precinct, Dutton Park, QLD 4102, Australia. [2] Office of Environment and Heritage, Sydney, NSW 2000, Australia. [3] Centre for Ecosystem Science, University of New South Wales, Kensington, NSW 2033, Australia. [4] INRA UR875 Unité de Mathématiques et Informatique Appliquées, 31320 Toulouse, Castanet Tolosan, France. Correspondence and requests for materials should be addressed to S.N. (email: sam.nicol@csiro.au)

Threats to biodiversity are numerous and increasing over time[1,2] resulting in growing lists of threatened species and ecosystems globally[3,4]. Resources available for species recovery are not keeping pace[5,6], precipitating an urgent need to ensure that conservation expenditure is used efficiently. Threat management can reduce the risk of extinction of threatened species, yet despite decades of research and considerable expenditure on threat abatement, the most effective actions for reducing threats are not always well understood[7,8]. This uncertainty costs management agencies, both in terms of wasted resources and missed opportunities to recover threatened species. Additional research to reduce uncertainty in threat management can be undertaken, but at a cost, and it is not clear where to direct limited funds to maximise return on investment from these studies[9]. Information will be most useful where it quantifies the impact of uncertainty about threatening processes on species persistence. Investment in uncertainty reduction must then target critical sources of uncertainty to maximise returns for the greatest number of threatened species.

The value of reducing uncertainty can be quantified using value of information (VOI) analysis, which evaluates whether collecting more data could lead to improved management outcomes[10,11]. The use of VOI to evaluate the expected utility of knowledge gain in conservation is growing[12–15] but has only recently started to consider multi-species, multi-threat prioritisation problems that are relevant to prioritisation of state-level or national-level threatened species lists[9,16]. For example, Bal, Tulloch[16] developed a technique to determine the relative value of different monitoring methods for multiple species and threats, but their analysis did not quantify the magnitude of change in species persistence. Without an understanding of the magnitude of change in persistence, decision makers have inadequate information to assess the risk of extinction. Here we develop an approach that quantifies both the likelihood and magnitude of gains to species persistence from threat management.

Building on existing studies, we develop a general VOI approach to calculate both the expected benefit (gain in threatened species persistence) of management given current levels of uncertainty, and the expected gain in benefit from removing uncertainty about management effectiveness of key threatening processes (KTPs; see the "Methods" section) on groups of listed species (hereafter 'species groups') that respond similarly to a KTP. The decision problem we analyse is whether or not to manage a KTP, given uncertainties about the effectiveness of management and about what would happen to species groups without management. We apply our work to KTPs in the Australian state of New South Wales (where KTPs are formally listed alongside threatened species under the *Biodiversity Conservation Act 2016*), but most of the KTPs are archetypes of threats that are widespread and relevant throughout the world (see Supplementary Tables 1 and 2). We hereafter refer to KTPs as 'threats'. We first examine which threats are the best candidates for management under current levels of uncertainty before determining which threats would yield greatest returns from removing uncertainty. Following this, we repeat the analysis to determine promising species groups that can be targeted for uncertainty reduction. Finally, we examine which uncertainty (i.e. uncertainty about threat reduction or uncertainty about species' responses to management) should be removed to maximise the persistence of groups affected by each threat.

Our findings demonstrate that removing uncertainty about management effectiveness could triple the expected gains in species persistence compared to management under current uncertainty alone. We also show that there are patterns in the kinds of threats that managers are most uncertain about: management of fire, invasive animals and a plant pathogen are most impeded by uncertainty, while management of invasive plants is least impacted. Finally, we demonstrate that uncertainty about species' response to threat reduction is the most beneficial type of uncertainty to reduce. Our results emphasise the tremendous importance of reducing uncertainty about species responses to management, and show that failure to consider management effectiveness wastes resources and impedes species recovery

## Results and Discussion

**Managing some threats is effective despite uncertainty.** Without investment in learning, the average expected gain in persistence (compared to doing nothing) from threat management was 0.033 per species (persistence is measured on a 0–1 scale, so this equates to an absolute gain of 3.3% per species; we express absolute changes in persistence using percentages for the remainder of the text). Species impacted by invasive plants were likely to have greatest gains in persistence as a result of management under current uncertainty, particularly exotic perennial grasses, lantana, bitou bush and African olive (Fig. 1). Managing

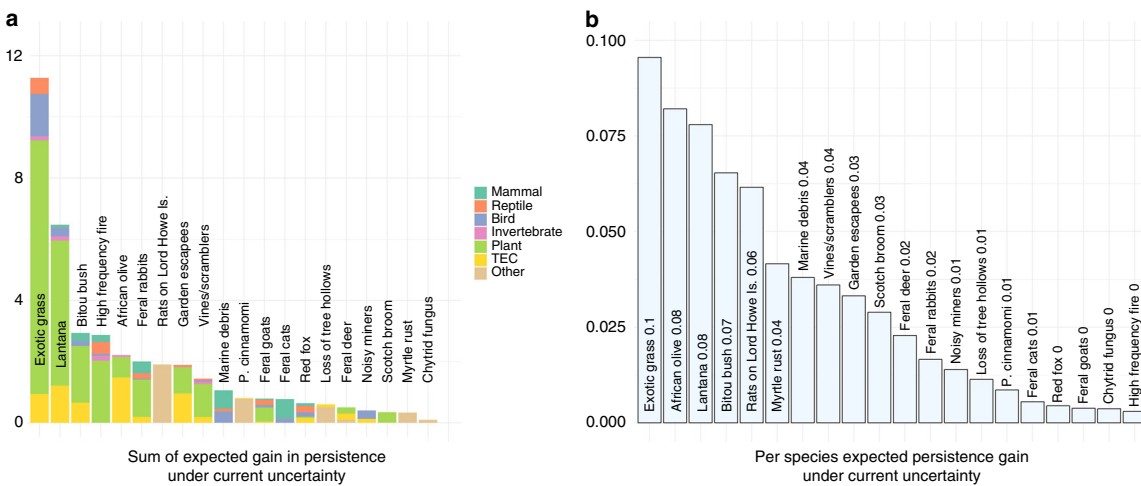

**Fig. 1** Expected gain in persistence from management for each threat under current uncertainty. Gain is computed relative to taking no action. Panel **a** depicts the total gain in persistence for all species affected by the threat; colours depict the proportional contribution to the sum from groups of affected species. Panel **b** depicts the mean per-species gain in persistence. Quantities in the bar labels are rounded to two decimal places

high-frequency fire had a high summed gain in persistence due to the extremely large number of listed species impacted by fire (972 impacted species and communities), however, the average gain in persistence of a single species from managing fire was negligible. Current threat management strategies for feral goats, cats and red foxes performed well for some well-understood species groups (e.g. critical weight range mammals[17]), but were not demonstrably better than doing nothing when averaged over all species likely to be affected by the threat[18–20]. This was due to extreme uncertainty about species responses in a number of bird, bat and reptile species groups.

**Resolving threat uncertainty may triple gains in persistence.** The value of removing uncertainty was measured using the expected value of perfect information (EVPI), which is the expected additional gain in persistence that could be achieved by removing uncertainty about whether or not to manage a threat, compared to taking the best action under uncertainty (see the "Methods" section).

For most threats, removing uncertainty about management efficacy and species response to threat reduction was likely to improve species persistence (Fig. 2). When management uncertainty was resolved, the order of the threats ranked according to persistence improvement was roughly the reverse of their ranking when ordered by persistence improvement from management under current uncertainty. Well-understood threats with high expected gains in persistence had little to gain from reducing uncertainty. Poorly understood threats were good candidates for uncertainty reduction.

When summed across all affected species, the total gain in persistence from removing uncertainty about the impacts of high-frequency fire was far greater than for any other threat (Fig. 2). The magnitude of the EVPI for this threat was driven by both expert uncertainty about the effectiveness of managing fire and the large number of species impacted by high-frequency fire. Although the mean predicted increase in persistence from fire management for most species groups was negligible, expert uncertainty about of the impacts of fire management (both the effectiveness of management and species responses) was severe[21,22], resulting in a high expected VOI. Although the value of removing uncertainty from so many species is potentially very large, the number of interacting factors, differing species responses and other practical considerations make reducing uncertainty about fires notoriously challenging[21,22].

The mean expected gain in persistence of removing uncertainty was 9.3% per species. Gain was highest for species impacted by high-frequency fire (14.8% per species) and the plant pathogen *Phytophthora cinnamomi* (14.4% per species), both of which affect many threatened species and have uncertain management effectiveness[21,23]. These gains were considerable, both in an absolute sense and when compared to the gains of managing using current uncertainty. Indeed, the average gain in persistence from removing uncertainty from a threat (12.6% increase per species compared to no action) exceeded the gains from managing under current uncertainty (3.3% increase per species). On average, the gains in persistence from removing uncertainty about management success would be expected to more than triple the gains expected from managing under existing uncertainty.

**Managing some species groups is effective despite uncertainty.** Understanding which threats offer the highest average gains in persistence is useful for large-scale goal setting, but the process of averaging across species groups hides useful information about variability between affected groups. We analysed species groups to better understand how affected groups may respond to threat management and to identify opportunities for management and learning (Fig. 3 demonstrates the findings of this analysis using a single threat as an example).

Management under current uncertainty had the potential to substantially improve the persistence of some species groups when compared to doing nothing. The persistence of 11 species groups could potentially be increased by an average of at least 10% per species, and the persistence of a further 46 species groups could be increased by an average of at least 5% per species. With the exception of two groups (resilient plant species affected by high-frequency fire; species affected by competition with rats on Lord Howe Island), the 10 species groups with highest expected benefit under current management were affected by an invasive plant threat (Fig. 4a, b). In particular, species groups impacted by exotic perennial grasses made up roughly half (4/10 or 6/10) of these groups (Fig. 4; for results for all species groups see Supplementary Figs. 1 and 2). Both management of invasive plant threats and the responses of threatened species to plant threat reduction were comparatively well understood—for these threats,

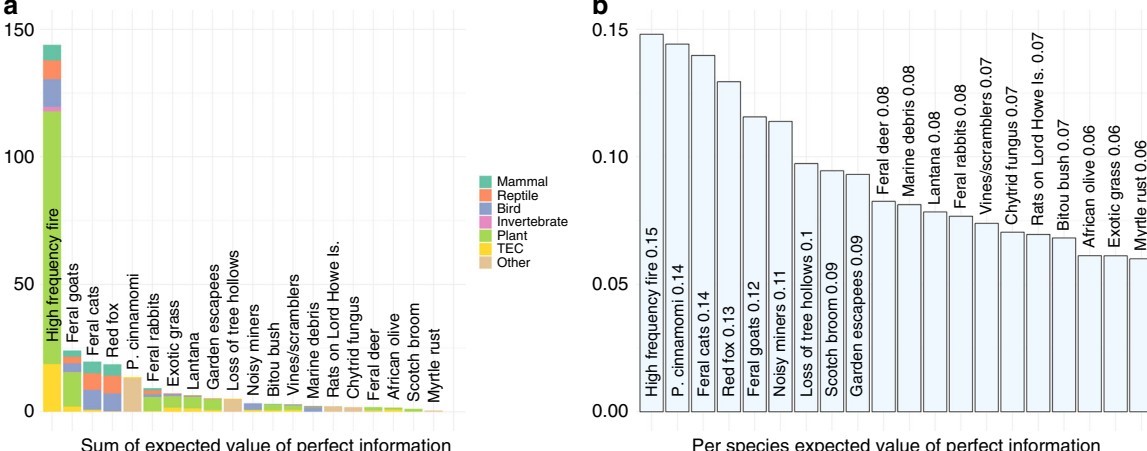

**Fig. 2** Expected value of perfect information (EVPI) for each threat if uncertainty about management effectiveness was removed. Panel **a** depicts the total additional gain in persistence for all species affected by the threat compared to the benefit of managing under current uncertainty; colours depict the proportional contribution to the sum from groups of affected species. Panel **b** depicts the per-species additional gain in persistence, compared to the per-species benefit of managing under current uncertainty. Quantities in the bar labels are rounded to two decimal places

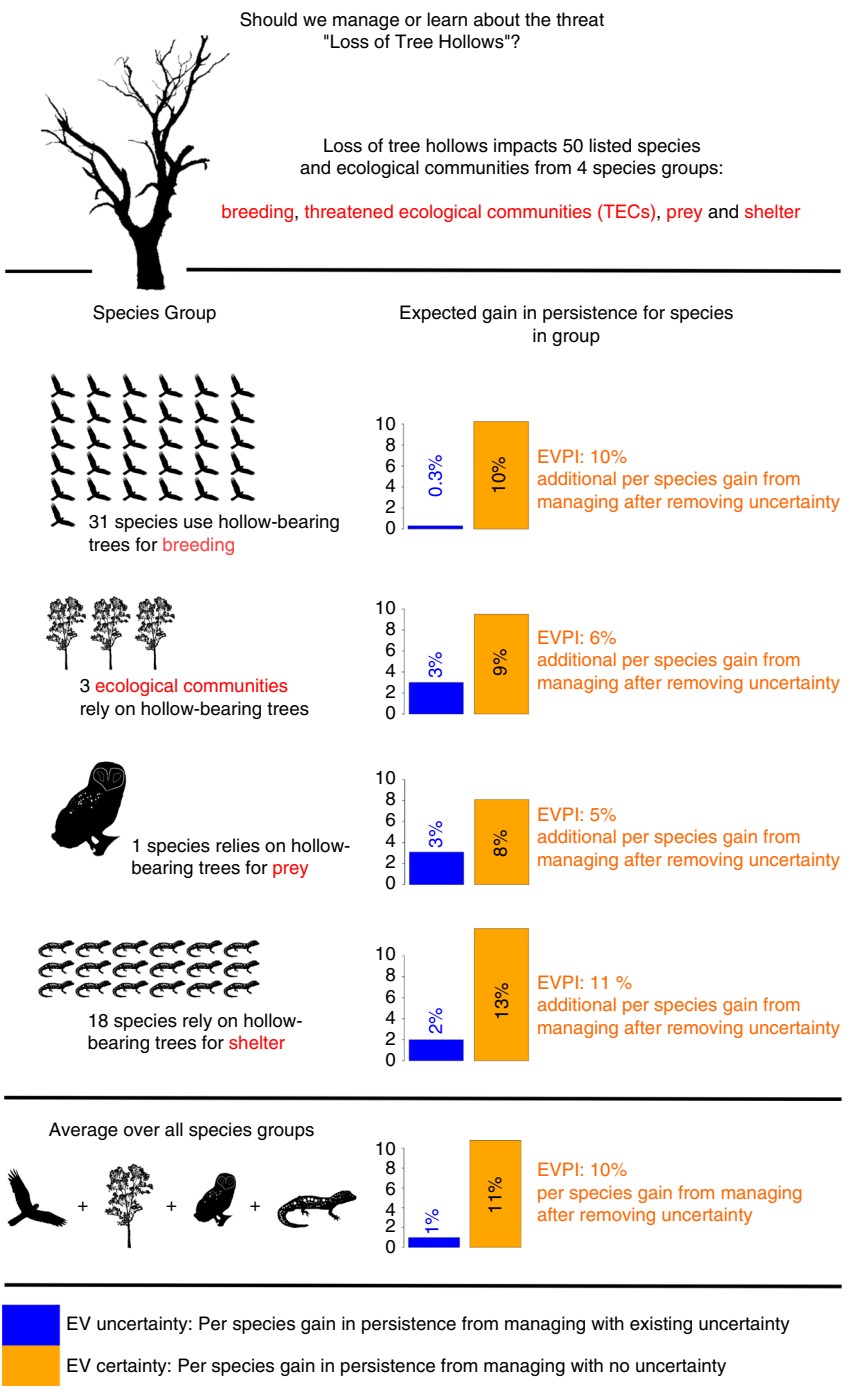

**Fig. 3** Interpreting the EVPI analysis for a threat. The graphic shows the results we obtain from our value of information analysis for the threat 'Loss of tree hollows'. The bar charts in the graphic compare the expected absolute gain in persistence under current uncertainty (EV uncertainty) and the expected gain in persistence if uncertainty was removed (EV certainty). The expected value of removing uncertainty (EVPI) is the difference between EV certainty and EV uncertainty. Values are computed for each affected species group and the average over all species groups affected by the threat

implementing best management practice using current knowledge may be a good strategy.

**Prioritise resolving uncertainties about predators and fire.**
Persistence of many species groups were unlikely to be improved with management under uncertainty, compared to doing nothing (62 groups had predicted per-species increases of <1%). The groups that were unlikely to be improved by ongoing management generally fit into two opposing classes: relatively high

expected persistence with high confidence without management (i.e. difficult to improve persistence further because it is already high), or total uncertainty about outcomes (i.e. the outcomes of management are too uncertain to predict the outcome in terms of species persistence). The latter group is likely to benefit from investment in uncertainty reduction.

The species groups with highest additional gain in persistence from removing uncertainty were dominated by high-frequency fire and feral animal threats. Experts were highly uncertain about the impacts of high-frequency fire on the majority of species

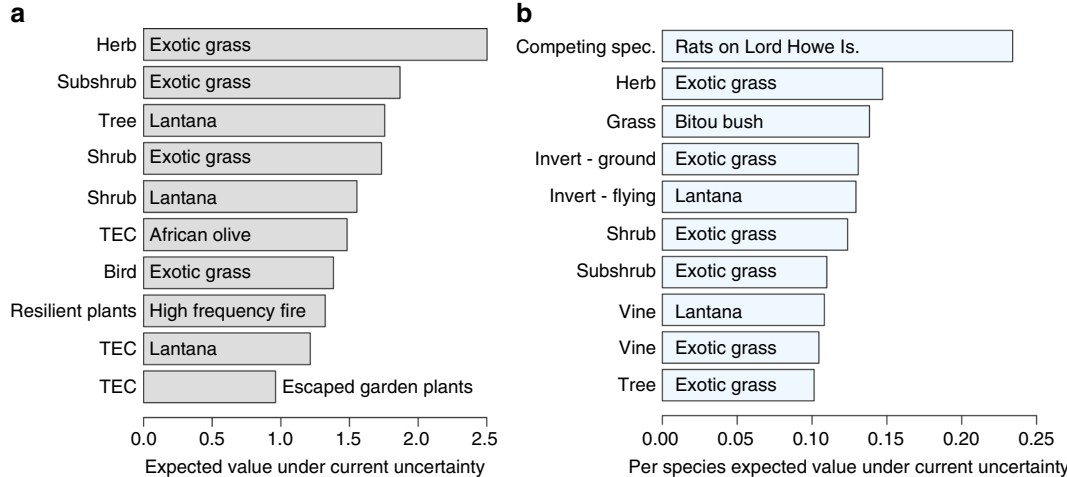

**Fig. 4** Expected gain in persistence for the 10 species groups that would gain most from threat management under current uncertainty. Gain is measured relative to no management. Plots **a**, **b** show the species groups with highest total gain in persistence and per-species gains, respectively. In each plot, the y-axis lists the species group that is impacted by the threat listed within the bars. For example, in **a**, the species group that has highest expected value from management is Herbs (species group) affected by Exotic grasses (threat). Note that the abbreviation 'TEC' stands for threatened ecological community

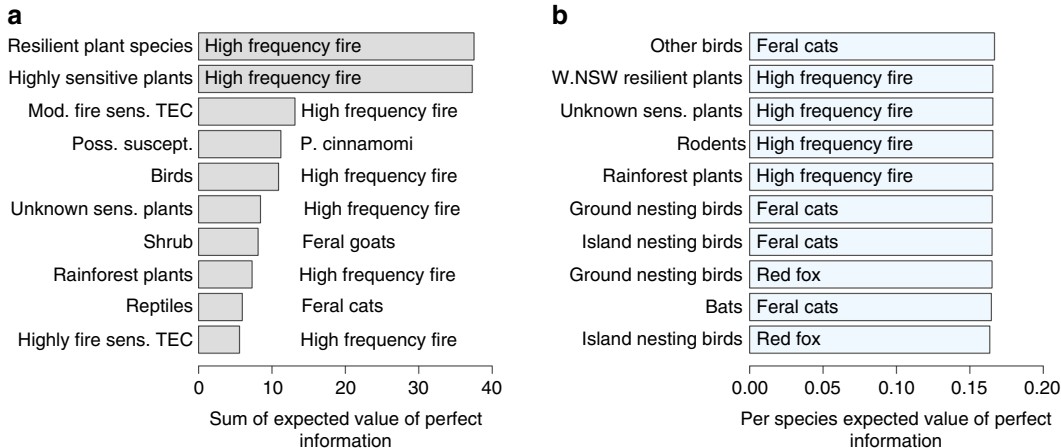

**Fig. 5** Expected value of perfect information (EVPI) for the 10 species groups with highest additional gain from threat management if uncertainty about management effectiveness was removed. Additional gain is measured relative to best management under current uncertainty. Plots **a**, **b** show the 10 species groups with highest total EVPI and EVPI per species, respectively. In each plot, the y-axis lists the species group that is impacted by the threat listed within the bars. Note that the abbreviation 'TEC' stands for threatened ecological community

groups considered[21,22]. This uncertainty had a particularly severe impact when summed over large species groups. In particular, two species groups, highly sensitive and resilient plant species (254 and 269 species, respectively; Fig. 5a) had the highest total EVPI and together accounted for half (51%) of the EVPI for the high-frequency fire threat. Experts were also uncertain about the impacts of foxes and cats on a number of bird, bat and reptile species groups[18–20]. Eliminating these uncertainties would potentially lead to substantial improvements in persistence compared to managing under current uncertainty. For example, eliminating uncertainty about the impacts of feral cats on three bird groups could lead to an additional 16% per species improvement in persistence for the species in these groups compared to managing under current uncertainty (Fig. 5).

Across all the threats, animal species groups tended to have more to gain from removing uncertainty than species groups of plants and ecological communities (Supplementary Figs. 3 and 4). There could be substantial improvements in the persistence of affected threatened species if uncertainty about invasive animal impacts was reduced or removed. High-frequency fire was an exception to this trend; both plant and animal groups could gain considerably from removing uncertainty about the impacts of fire.

**Species' response to management has high information value.** The expected value of partial information (EVPXI) represents the improvement on the gain in persistence if only some components of uncertainty are resolved. We examined the impacts of resolving three components of uncertainty. Specifically, we examined the impact of resolving uncertainty about the effectiveness of threat management actions, the species response to management, and the species response if no management was implemented.

Removing uncertainty about the effectiveness of management actions to control a threat had very little EVPXI value (Fig. 6) because doing nothing always had zero effectiveness. Taking an action was always at least as good as doing nothing, so in the absence of other information, the best decision was always to

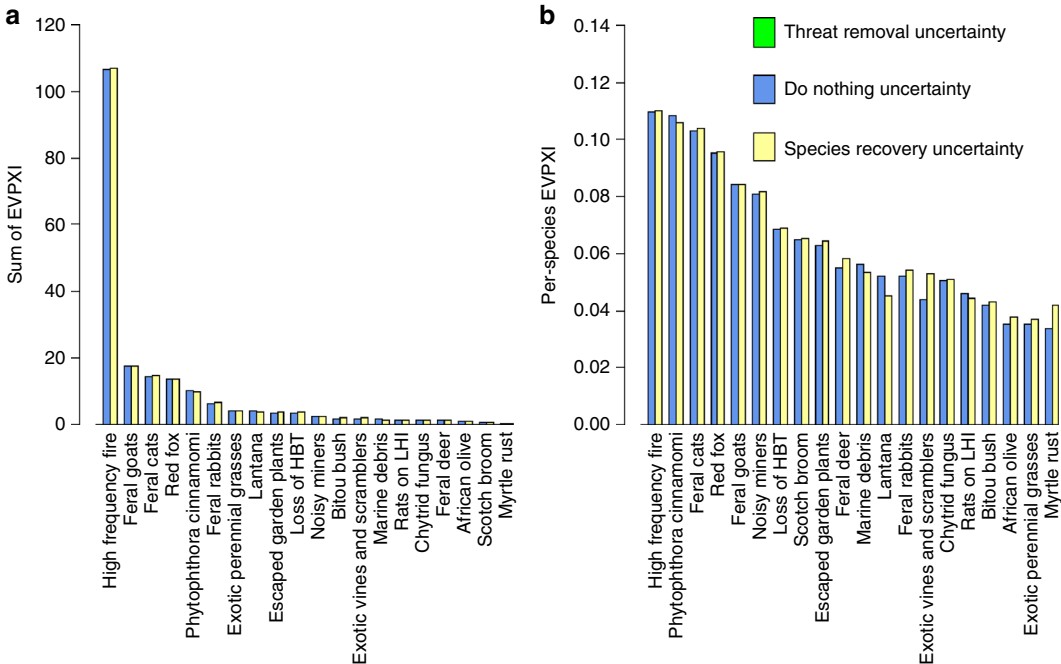

**Fig. 6** Cumulative expected value of partial information (EVPXI) for each threat. Colours indicate the contribution of each type of uncertainty; the length of the stacked bars indicates the size of the EVPXI for each uncertainty. Panel **a** depicts the average total additional gain in persistence for all species affected by the threat compared to the benefit of best management under uncertainty. Panel **b** depicts the per-species gain in persistence. In this study, reducing uncertainty about threat removal (green in the figure legend) has negligible value of information so does not appear in the plot

manage (assuming the cost of management could be borne and that management was feasible). In other words, it is sufficient to know that managing is better than doing nothing—the knowledge about the exact management effectiveness had very low value. Importantly, this is not the same as suggesting that improving threat management has no value. The EVPXI measures the benefit of partially reducing uncertainty relative to a baseline of best management. Improvements in management effectiveness would improve the baseline, but reducing uncertainty about management effectiveness would not change the decision about whether or not to manage a threat. Our research does not address the future benefits of studies that could improve on the current best-practice management actions evaluated in our study. In addition, the low value of learning about the effectiveness of threat management actions is partly an artefact of our EVPI (and EVPXI) formulation, which considers only two actions: manage the threat or do nothing. If, instead of comparing managing relative to doing nothing, we compare managing either of two different threats, there may be greater value in knowing the relative effectiveness of each threat management strategy. The EVPI for any pair of threats can be computed—see the subsection 'General case: two key threatening processes' in the "Methods" section for the formulation.

In contrast, the species response to threat removal can be better or worse than doing nothing. In some cases, doing nothing may be better than taking action. While managers should always manage in order to reduce a threat, whether or not to manage to prompt a species response is uncertain.

Learning about the expected species response has considerable information value, either with management or in the absence of management (Fig. 6). Although our results suggested that learning about the response to management was slightly more beneficial than learning about response in the absence of management for 15 of 20 threats, the difference in value is negligible (mean increase in persistence of 6% per species both

with and without management). This demonstrates the considerable importance of understanding the counterfactual—i.e. knowing what would have happened if no management was undertaken.

**Using VOI for resource allocation decisions**. Some general trends arose from the analysis that can help to predict where to invest in reducing uncertainty for better threatened species outcomes. Learning about the effectiveness of management in removing threats generally had limited influence on species outcomes; most of the observed benefits could be obtained by learning about the difference between species persistence distributions with and without management. To predict which threats or species groups would benefit most from learning, the key factors were the amount of overlap between the distributions of persistence with and without management and the variability in the distributions. Where there was little overlap or the distributions were very tight, there was no uncertainty about whether management was better than doing nothing and therefore no need to invest in learning—these species groups were good candidates for ongoing management. In contrast, where there was considerable or total overlap between the distributions with and without management, and high variability in the distributions, there was considerable uncertainty about whether management was better than doing nothing—as a rule of thumb, these species groups were good candidates for research to reduce the uncertainty about management responses. The very best candidates for learning were those which had uniform distributions (i.e. full overlap and maximum variability) for the do nothing and/or managed persistence distributions.

In some cases the shape of response functions also influenced the expected benefit of taking action. For example, eradicating rats from Lord Howe Island had some of the highest predicted increases in persistence across all species groups under current uncertainty. However, the response functions for species groups affected by this

threat were extremely steep and resembled step functions. Only once management effectiveness became very high (>80–96.25% for all but one one species group affected by this threat) was it possible to improve persistence by taking management action. This requirement for extremely high effectiveness is likely because rats need to be eradicated from the island or they will recolonise.

Our decision problem compared the value of removing uncertainty about whether to manage a threat or not. The EVPI is useful for evaluating the absolute additional gain in persistence that could be obtained by removing uncertainty about management effectiveness, in comparison with the counterfactual, for each threat individually. However, our results do not evaluate the relative value of resolving uncertainty about managing two threats in comparison to each other. Resolving which of two management actions to take is a different decision problem that is best solved on a case-by-case basis due to the large number of possible pairwise combinations of threats (see the subsection 'General case: two key threatening processes' in the "Methods" section for a procedure to compute this).

Our study quantifies the expected gain to threatened species persistence when removing uncertainty about threat management effectiveness, providing decision-makers with useful information to efficiently prioritise limited resources. However, other factors are also important to consider when allocating funds. In particular, combining our findings with the costs of information gain will determine the most efficient allocation of funds. There are well-developed methods to cost-effectively allocate limited resources to manage threats[8], which could be used in conjunction with our methods to make cost-effective decisions for threat management. If the costs are known, proposed adaptive management studies to reduce uncertainty can be evaluated by comparing the cost of research with the VOI[24].

A second factor to consider is interactions between threats. Threat interactions were not captured by our method due to the large number of possible combinations and a lack of information about relative threat strengths. Quantifying the VOI with interacting threats is difficult because it requires experts to parameterise a joint distribution that characterises the extent to which the intensities of interacting threats modify the intensity and benefit of managing a target threat. As a proxy, we considered the extent and frequency of interactions in our raw data set (Fig. 7). In our study, threat interactions were dominated by high-frequency fire (Fig. 7a). Most species impacted by a threat were also impacted by fire (mean interaction frequency was 78%). Species impacted by invasive animals (rabbits, foxes, goats, pigs and cats) also shared threats moderately frequently, but other interactions were relatively rare (Fig. 7b; excluding interactions with fire, the mean proportion of species that shared two threats was 7%). While interactions between threats are likely to be important for species impacted by fire and are undoubtedly critical for particular species with known strong interactions (such the well-documented impacts of mesopredator hyperpredation and prey switching on small mammals in Australia[25]), the lack of interactions in our analysis suggests that there may be some threats that can be managed effectively in isolation. Threats with fewer interactions correlated roughly with our findings about the best threats to manage under current uncertainty, suggesting that management is more effective when fewer threats interact. Threats with many interactions aligned roughly with the threats with high VOI, suggesting a possible link between the number of threat interactions and the impact of uncertainty on management effectiveness.

Some results of this study may be place-specific or expert-specific. Our study elicited data from a particular set of experts, and it is possible that a different set of experts would assign different benefits during elicitation. Indeed there is evidence that the importance of threats vary in different parts of the world[26], and this may make different expert rankings valid in different places. Our findings nonetheless represent the largest survey of threat importance and the VOI to date and are likely to be useful for areas that face similar threats to New South Wales, and our methods pave the way for similar surveys to be conducted in other jurisdictions.

A limitation of elicitation in threatened species management is that sometimes few experts are available. For example, here the expected impacts of high-frequency fire are obtained from one

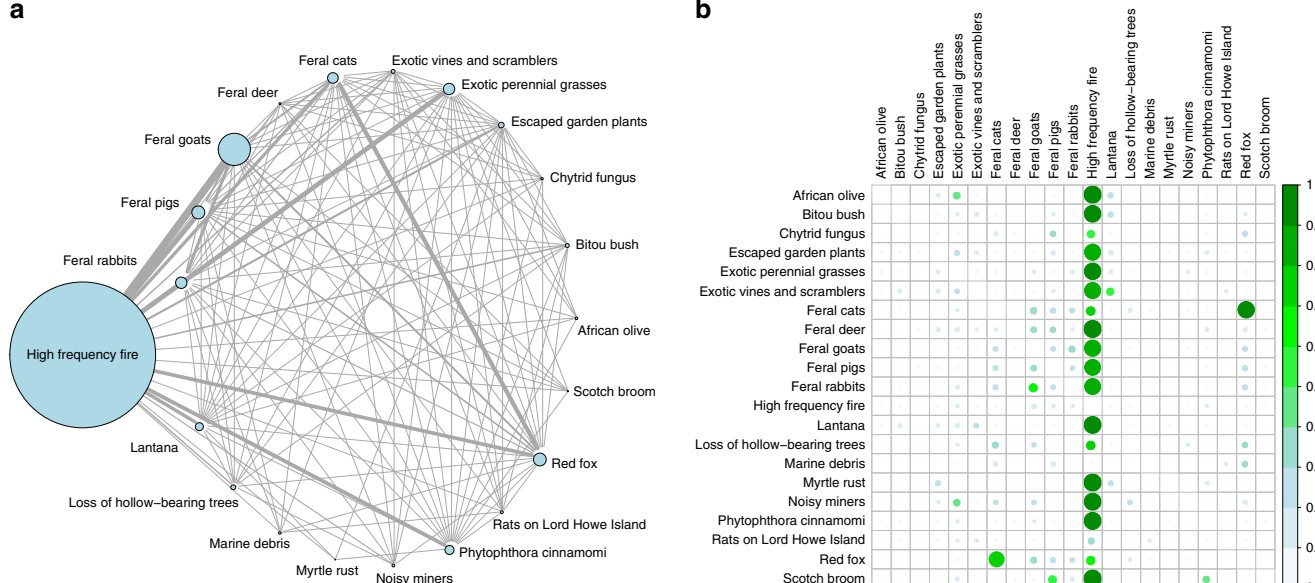

**Fig. 7** Extent and frequency of interactions between key threatening processes. Panel **a** illustrates the number of interactions between threats. Network edge sizes are scaled to the number of species that are affected by a pair of threats; node sizes and colours are scaled to the number of species affected by the threat. Panel **b** illustrates the frequency of the interactions. Rows show the proportion of species affected by the threat that are also affected by the threat in each column

expert (see Supplementary Table 2). In the absence of additional information, it is better to use the expert information available and critically examine the results than to ignore the threat. For example, here the findings of high VOI for the fire threat are likely to be robust, as: (i) most of the potential value of fire management comes from the sheer number of species affected by fire rather than from expert-predicted benefits of management (the predicted gains from management were very low for most species groups affected by fire), and (ii) the uncertainty which creates the VOI is well documented for fire[21,22].

Our study has three clear findings. Firstly, we have quantified the substantial impediment that uncertainty poses to effective management of key threatening processes. We showed that research to remove uncertainty in conjunction with existing management could triple the expected gains in species persistence compared to management under current uncertainty alone. While other studies have noted the value of gaining new information for conservation purposes before action[27–29], none have yet quantified the value for reducing such a broad range of threats, nor found such a strong effect. Secondly, we discovered that uncertainty about species' response to threat reduction was by far the most beneficial type of uncertainty to reduce. There is a tremendous need to improve data collection about species' responses to management globally[30], and our study both quantifies the cost of our collective ambivalence towards recording the effectiveness of conservation interventions and identifies which uncertainty to study to most benefit threatened species. Finally, we showed that there were strong patterns in the kind of threats that managers are most uncertain about. In general, invasive plant management and island eradications are well understood and are good candidates for management without considerable new investment to reduce uncertainty about effectiveness, but the impacts of fire, invasive animals and a plant pathogen on many groups of threatened species are poorly understood and may be good candidates for future research[18–20,23]. Despite the potential for regional variation in the importance of uncertainty about specific threats, the threats in our study can be considered as archetypes of global threats to biodiversity, and commonalities in management actions for archetypical threats across the globe make it probable that the general findings will be applicable to other locations. Our work is the largest study of its kind yet conducted and provides a benchmark method and results that can be built upon to inform management at any location to better prioritise the allocation of resources towards biodiversity research.

## Methods
**Method overview—synopsis.** The decision problem we considered was to choose whether to manage a given threat or not. This required evaluating the expected gain in persistence from managing a threat, compared to doing nothing, when the outcomes of both management and doing nothing were uncertain. To quantify the benefits of managing threats with and without uncertainty, we used an expert elicitation approach to evaluate the VOI for improving the management of 20 listed threats in New South Wales[31]. A total of 261 experts were invited to contribute by email, of which 66 provided estimates. 976 listed species and ecological communities were allocated to 60 species groups based on similar responses to threats and presented to experts (see Supplementary Table 2 for breakdown of listing categories). For each threat and species group, we elicited three pieces of information from experts: (1) the likelihood that best-practice management would effectively manage the threat; (2) the average probability that species groups would persist if no management was undertaken; and (3) the average probability that species groups would persist if best-practice management was applied. In each estimation, experts provided lower, upper and best guesses, as well as their confidence that the true value lay between the lower and upper estimates[32]. Estimates were then fitted with probability distributions representing the likelihood of effective management and the likelihood of species group response for a given level of management effectiveness. These distributions were used to calculate the expected gain in persistence for each species group resulting from threat management under (i) current uncertainty ($EV_{uncertainty} = \max(0, E(b_1 - b_0))$), where $b_1$ and $b_0$ are the probabilities of persistence resulting from managing a threat and doing nothing,

respectively) and (ii) perfect knowledge, i.e. if uncertainty about management outcomes was eliminated ($EV_{certainty} = E(\max(0, b_1 - b_0))$). The EVPI ($EVPI = EV_{certainty} - EV_{uncertainty}$) and expected value of partial perfect information (EVPXI) were then computed for each species group[12]; these quantified the likely additional gains from removing all or part of the uncertainty about management outcomes, respectively, compared to best management under uncertainty (see subsection 'Comparing the value of action to doing nothing' for details of calculations). The expected gain in persistence under current uncertainty, the EVPI and the EVPXI for each species group were aggregated to the threat level by summing the species group results. The summed gain in persistence is useful when the objective is to maximise gain in persistence, however this metric will be biased towards threats that affect many species, which may result in high overall gains caused by summing negligible gains over many species. An alternative objective is to maximise the per-species benefit so that affected species receive a significant gain in persistence as a result of management. The mean benefit per species was computed by dividing the summed result by the number of species affected by the threat.

The remainder of this section provides a full description of our methods.

**Case study: The Saving our Species (SoS) programme.** The New South Wales (NSW) government is investing $100 million over 5 years into the Office of Environment and Heritage (OEH) SoS programme, which aims to maximise the number of threatened species and ecological communities that are secured from extinction in the wild for 100 years and to control key threats facing threatened plants and animals. The programme has core principles of cost-effectiveness, scientific rigour and transparency which are being applied to guide investment in NSW. Already these have been applied to prioritise management of almost 450 site-managed species using a cost-effectiveness approach[33]. This process of prioritising threatened species projects led to the creation of a database of expert-derived data that is critical for VOI. For example, the SoS database contains estimates of the expected persistence of threatened species in the absence of management to mitigate threats.

Key threatening processes (KTPs) are drivers of extinction for species and ecological communities. Managing KTPs that affect many species can be more cost-effective than managing species individually[1]. Although there are threats other than KTPs that are important factors affecting listed species (such as grazing), KTPs are a legislative focus of the SoS programme. NSW currently has 38 listed KTPs, which are processes that adversely affect threatened species or ecological communities or could cause species or ecological communities to become threatened. Of these, OEH will develop KTP strategies only for those KTPs which critically impact threatened species or communities and for which targeted actions are likely to contribute significantly to the abatement of these impacts[1].

The areas impacted by KTPs in NSW are vast and KTPs impact over 1000 listed threatened species and communities. However, the funds to manage them are limited and many species remain unfunded despite the NSW government's considerable investment. In this situation, smart resource allocation decisions are necessary to maximise the number of species that benefit from management. In some cases the benefits of managing KTPs are only partially known because the outcomes of management cannot be easily disentangled from chance events. Removing this uncertainty about management effectiveness has the potential to improve KTP management outcomes, but investing resources in learning may detract from immediate management and therefore needs to be justified by an expected future improvement in management. Here we use VOI techniques to evaluate the expected increase in species persistence resulting from removal of this uncertainty, which can be used to determine whether or not to invest in reducing the uncertainty about the effectiveness of managing a KTP.

**Method overview—intuition.** Our method finds two key pieces of information: the expected gain in persistence that would be achieved by managing each KTP under current uncertainty (compared to doing nothing), and the expected additional gain in persistence that could be achieved by removing uncertainty about whether or not to manage a KTP (i.e. the VOI). For brevity, we hereafter refer to the expected gain in persistence as the 'benefit' (a glossary of key terms is included in Supplementary Table 3).

The first step in our process was to group species based on similar responses to a KTP. In the text we refer to these groups as 'species groups'. Individual species can be impacted by multiple KTPs, so the same species can be present in multiple species groups (Fig. 8).

After grouping species, our approach uses expert elicitation to estimate the average per-species gain in persistence from managing a species in a species group impacted by a KTP. To help experts make their estimates, we precomputed an estimate of the expected gain in persistence for species in each group as a result of managing a KTP (Supplementary Fig. 5; experts could overrule our estimate). In our estimate, the persistence of each listed species was assumed to be limited by a number of KTPs. The severity of KTPs varied by species. We assumed that removing the influence of a KTP by management would result in a gain in persistence that was proportional to the severity of the threat removed and the number and severity of the other KTPs affecting the species. We repeated this calculation for each species in the group then obtained the average estimated gain in persistence across all species in the group.

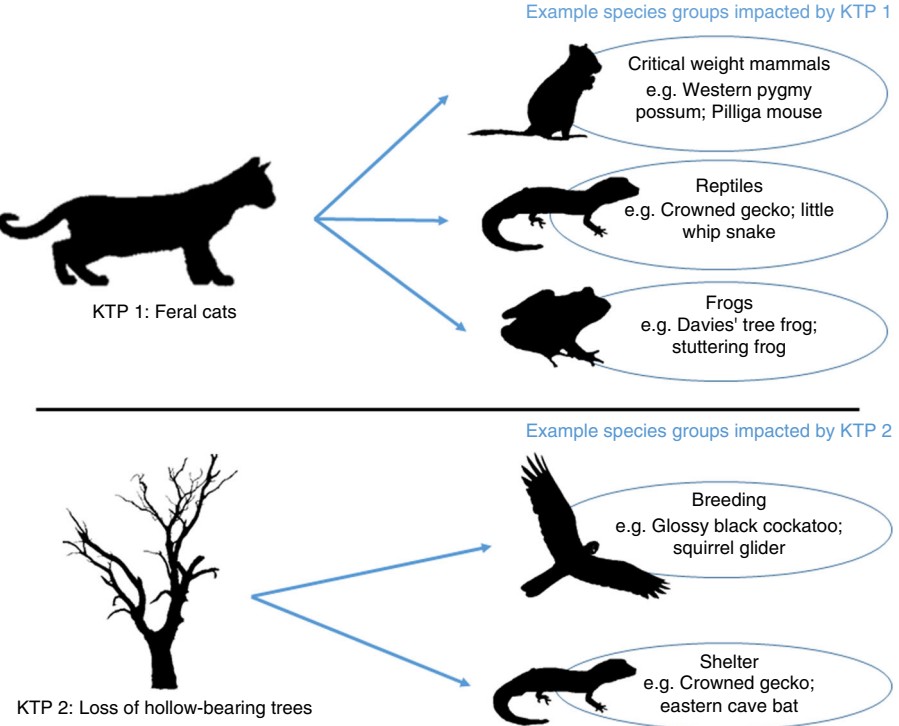

**Fig. 8** Schematic illustrating the relationship between KTPs, species groups and species for two example KTPs. Species groups are composed of species, which are impacted similarly by a particular KTP. Note that not all KTPs, species groups or species are shown in the schematic

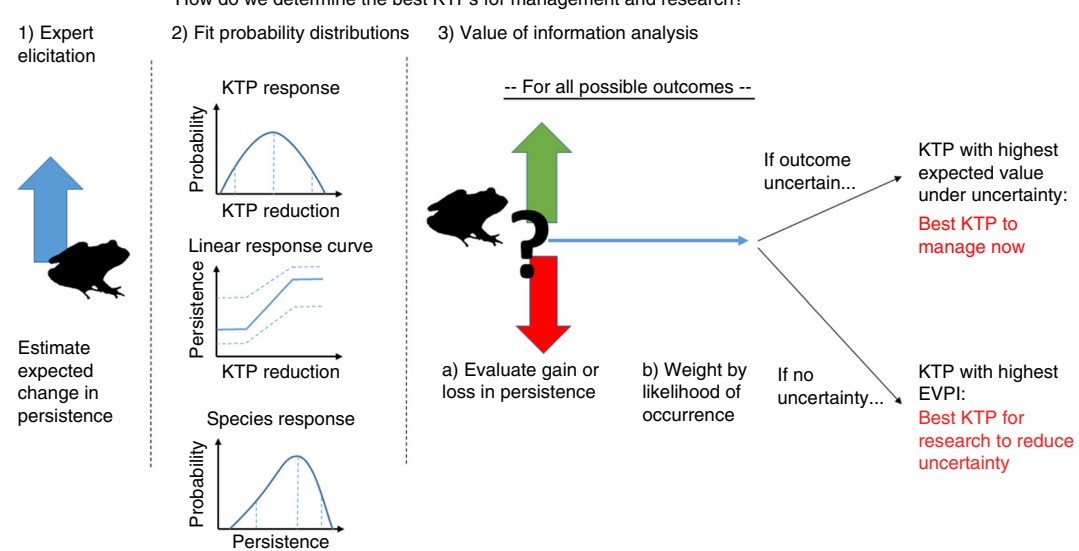

**Fig. 9** Schematic of the process used to determine the best KTPs for management and research

A schematic of the process used to estimate the benefit of managing a KTP is included in Fig. 9. Guided by the initial estimates provided (see previous paragraph), experts estimated how effective management is at removing a KTP and the persistence with and without management for each species group affected by a KTP. The expert estimates were used to parameterise probability distributions that described the probabilities of any possible persistence outcome. These distributions were averaged across experts to give a single distribution for each species group. We then used the distributions to compute the expected gain in persistence from management compared to doing nothing and the additional benefit of removing uncertainty compared to best management under uncertainty (i.e. the VOI).

With existing levels of uncertainty, this process gives us the expected gain in persistence for the species group if we were to act under current uncertainty

compared to doing nothing. We computed the expected gain in persistence under current uncertainty (and later the VOI) at two levels of aggregation: (i) at the species group level using the approach already described; and (ii) at the KTP level. To aggregate the expert estimates made at the species group level to the KTP level, we summed the expected gains in persistence for all species groups affected by the KTP. An average per-species gain in persistence for the KTP was obtained by dividing the sum by the number of species affected by the KTP. The KTP that gives the highest expected gain in persistence is the best KTP for management with existing levels of uncertainty.

If there is no uncertainty, we can choose the best management action before the outcome of management is realised—in this case we have perfect information. The expected gain in persistence that could be obtained from removing uncertainty is the VOI; the KTP with highest VOI is the best choice for uncertainty reduction.

In the following sections we elaborate on this process. We first introduce the concepts of the two main pieces of information that we obtain from the analysis: (i) expected benefit under current uncertainty and (ii) VOI. Note that the VOI calculation requires the expected benefit under current uncertainty, so we need to compute (i) before we can compute (ii).

**Expected benefit under current uncertainty**. Under current uncertainty, the best decision is determined by the utility of management outcomes and their likelihood of occurrence. The best management action is the one which has highest expected utility; this maximum expected utility is hereafter referred to as $EV_{uncertainty}$. Formally:

$$EV_{uncertainty} = \max_{a \in A} \left[ \int_{s \in S} u(a,s) f(s) ds \right] \quad (1)$$

where $a \in A$ is a management action, $u(a,s)$ is the utility (in this case, gain in persistence) of obtaining outcome $s$ after taking action $a$, and $f(s)$ is the probability density distribution representing prior belief about the likelihood of an outcome $s$. In our case we consider two actions per KTP, corresponding to best practice management of the KTP and taking no action.

**Value of information**. VOI analysis provides a framework to quantitatively estimate the value of acquiring additional evidence to inform a decision problem. In this study, we used VOI analysis to determine the value of resolving uncertainty on the efficacy of threat reduction and on the species response.

The VOI is the difference between the expected management outcomes (in our case, in terms of improved species' persistence) when a decision is made only on the basis of the prior information and the expected management outcomes when new information is gained[11,12]. In the case where the new information eliminates uncertainty entirely, the resulting expected utility is called the EVPI. In plain language, EVPI asks: "how much better could my management outcomes be if all uncertainty was removed?" Formally:

$$EVPI = EV_{certainty} - EV_{uncertainty}$$
$$= \int_{s \in S} \left[ \max_{a \in A} u(a,s) \right] f(s) ds - \max_{a \in A} \left[ \int_{s \in S} u(a,s) f(s) (ds) \right] \quad (2)$$

In our case we consider two actions per KTP, corresponding to best practice management of the KTP and taking no action. The first term in Eq. (2) represents the expected value of the action maximising the utility $u$ with no uncertainty— i.e. when the decision-maker can choose the best action for each value of $s$. The second term in Eq. (2) represents value of the action maximising the utility when taking into account the uncertainty on $s$—here the decision-maker does not know the outcomes of the actions in advance and so chooses the option which maximises the expected value[12].

**Management objective and benefit metric**. The management objective is to select actions that maximise the improvement in the probability of persistence of listed species over the next 100 years. A threatened species is defined to persist if there is a 95% probability of having a viable population of the species in 100 years from now, and the species' threat status under the Threatened Species Conservation Act will not decline[34]. We measure the probability of persistence as the likelihood that the species will persist according to this definition. The likelihood of persistence and other data for the project were generated using expert elicitation and background information from threat and species databases (see subsection 'Data collection').

For a species group, our benefit metric ($u$ in Eqs. (1) and (2)) is the change in the probability of persistence of a 'typical' species within a species group as a result of the management action. For a KTP, the benefit metric is a weighted average of the species group benefits for all groups affected by the KTP as a result of the management action, where weights are determined according to the number of species in each species group.

In this study, we defined the benefit metric as $u(i) = (b_i - b_{i0})$, where $b_i$ is the expected probability of persistence of a typical species from the group (or KTP, depending on the level of aggregation) after taking management action to reduce threat $i$ and $b_{i0}$ is the expected persistence if no management action is taken. It is also possible to define the benefit metric to compare two different KTP management actions (a formulation for this is given in subsection 'General case: two key threatening processes' below), but due to difficulties summarising all pairwise comparisons, we did not use this approach here.

Our benefit formulation generates the expected per-species gain in persistence as a result of management action. This metric is useful for ensuring management is not biased towards large species groups. In our results we also report the total gain in persistence of the species group or KTP by multiplying the per-species gain in persistence by the number of affected species. This metric is useful for determining the total gain in persistence from managing a species group or KTP.

**Uncertainty model**. The uncertainty model ($f(s)$ in Eqs. (1) and (2)) consists of two components—the extent to which best-practice threat management can reduce the severity or extent of the KTP, and the species response (change in probability of persistence of species) given a reduction in threat. Concretely these two sources of uncertainty are quantified by $\theta_i$, the level of management effectiveness reported as a proportion of the initial threat removed after management ($\theta_i \in (0, 1)$), and $b_i \in (0,1)$ the probability of persistence of the species after applying action $i$. In the following, $\theta_i$ will be called effectiveness and $b_i$ will be called persistence.

The joint probability of these two events can be specified as

$$Pr(b_i, \theta_i) = Pr(b_i | \theta_i) Pr(\theta_i) \quad (3)$$

$Pr(\theta_i)$ is the probability of achieving effectiveness $\theta_i$ by applying management action $i$. $Pr(b_i | \theta_i)$ is the probability of achieving a persistence of magnitude $b_i$ by applying management action $i$, given that the effectiveness for this KTP is $\theta_i$.

The probability of a species response to a KTP reduction $Pr(b_i | \theta_i)$ is challenging to specify because it depends on the effectiveness of the KTP reduction $\theta_i$. Specifying species response at different levels of $\theta_i$ for every species would require impractical amounts of data. To reduce the amount of data to collect we grouped species into species groups based on assumed similar responses to KTPs. A response curve can be elicited for the whole species group rather than individual species. Response curves are functions that describe how the species response varies for any level of threat reduction. For ease of elicitation, we assumed piecewise linear response functions with fixed minimum and maximum management effectiveness values. Linear response functions assume that benefit is received in proportion to the effectiveness of the KTP. The response functions were generated from experts by eliciting the expected persistence with and without effective KTP management, as well as the minimum level of effectiveness required before persistence of the species group begins to increase and the level of effectiveness required to achieve maximum persistence of the species in the species group. Values of $Pr(b_i | \theta_i)$ can be interpolated from the response function (Fig. 10).

**Data collection**. Groups of listed threatened species affected by KTPs were collated by OEH staff, who referred to the NSW Threatened Species Profile Database and the species contained in each KTP listing to help assign species to KTPs. Species could be assigned to multiple KTPs. OEH staff then assigned species affected by a KTP to species groups based on their knowledge of the species, which were grouped based on similar responses to the KTP (Supplementary Table 1). Of the 38 listed KTPs, we selected 21 KTPs which were management priorities for OEH and adequately well understood to undertake our analysis.

Experts were selected based on recommendations from OEH staff and their knowledge of KTPs or the threatened species impacted by KTPs. Experts were predominantly ecologists or threatened species managers.

A total of 261 experts were invited to contribute by email, of which 66 provided estimates. A summary of the number of experts contributing to each KTP is contained in Supplementary Table 2. Although we received estimates for species group responses for the feral pigs KTP, we did not receive estimates of management effectiveness for feral pigs, so this KTP was removed from the analysis in this manuscript. Details of species included in each species group can be found in the data archive for this manuscript at: https://doi.org/10.6084/m9.figshare.7623665 (see folder: './Data_sets/KTPxFG_persistence.xlsx').

Experts were asked to provide parameters for the probability distributions in Eq. (3) based on the species groupings. The elicitation followed a four-point modified Delphi approach[32]. Experts were asked to provide a lower estimate (worst-case scenario), an upper estimate (best-case scenario), a best guess (most likely scenario), and the level of confidence that the true effectiveness lay between the upper and lower estimates. Mathematical consistency was checked automatically using an Excel macro to ensure that the confidence was greater than the difference between the upper and lower estimates (see data archive file: https://doi.org/10.6084/m9.figshare.7623665; see folder: './KTP EVPI project handover/Data_sets/KTP elicitation sheets/').

Experts provided estimates in two parts corresponding to the two sources of uncertainty, i.e. the effectiveness of threat management of the KTP (i.e. what was the likelihood of eliminating a proportion of the threat by best-practice threat management?) and the probabilities of persistence of species within the species group with and without management.

To elicit the first source of uncertainty, we defined the likelihood that management could eliminate a proportion of the threat by best-practice threat management ($Pr(\theta_i)$) as the extent to which management removed the KTP relative to threat levels if no management action was taken. 100% management effectiveness was defined as fully removing the threat. 0% effectiveness of management was defined as being equivalent to taking no action, such that the threat persisted at current levels or got worse. The management effectiveness estimates were used to parameterise a beta distribution for $Pr(\theta_i)$.

To elicit the second source of uncertainty, the likelihood that species respond to KTP reduction, the elicitation was organised in two parts. Firstly, we elicited the shape of the response functions for each species group. Experts were also asked to provide the minimum effectiveness of KTP management required before persistence of species within the species group begins to increase ($\eta$ in Fig. 10) and the level of effectiveness required before the KTP no longer impacts populations of the species in the species group ($\mu$ in Fig. 10). For both questions a qualitative scale was provided in the Excel spreadsheet (see data archive file: https://doi.org/10.6084/m9.figshare.7623665; see folder '/KTP EVPI project handover/Data_sets/

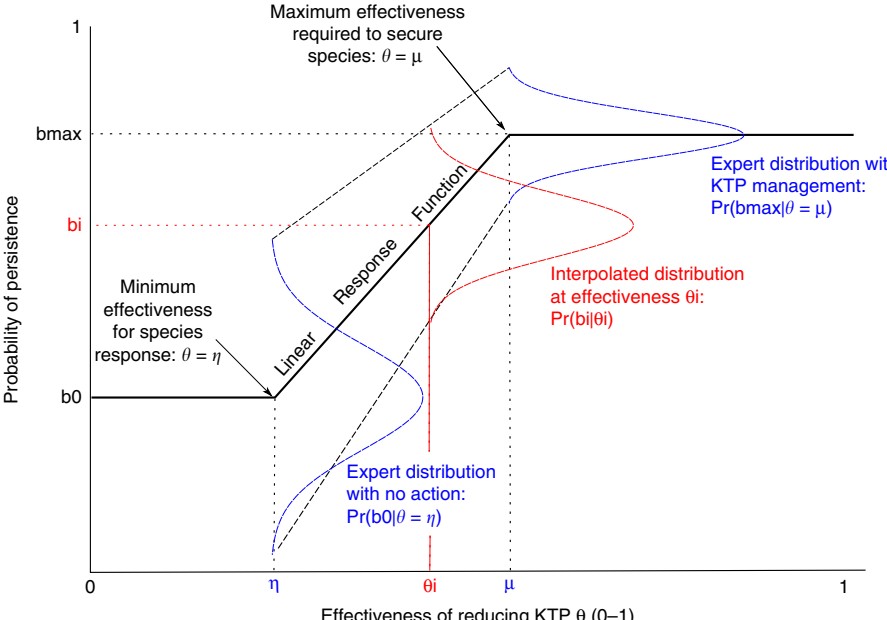

**Fig. 10** Illustration of how species group response curves $\Pr(b_i|\theta_i)$ are interpolated for varying levels of threat management effectiveness. The piecewise linear response curve is the solid black line, which is generated from expert estimates. The minimum level of effectiveness required before persistence of the species group increases and the level of effectiveness required to achieve maximum persistence of the species are located at $\theta = \eta$ and $\theta = \mu$, respectively. Experts provide four-point estimates that parameterise beta distributions when effectiveness is 0 and 1; these distributions are illustrated in blue in the figure. For values of management effectiveness between $\eta$ and $\mu$, the four-point estimates given by the experts are linearly interpolated according to the relative distance of the desired point from the extremes estimated by the experts (e.g. for the mode, interpolation is between $(\eta, b_0)$ and $(\mu, b_{max})$). The interpolated data are then used to fit $\Pr(b_i|\theta_i)$ at the desired management effectiveness (red curve in the figure)

KTP elicitation sheets/(KTPname).xlsx'). Secondly, we elicited the expected persistence of species within the species group after 100 years with and without management (lower, upper, best guesses and confidence values). The probabilities of persistence of the species group with and without management were used to fit distributions of $\Pr(b_i|\theta_i = 0)$ and $\Pr(b_i|\theta_i = 1)$, respectively. These two distributions were linearly interpolated between $\eta$ and $\mu$ using the response function described in Fig. 10. Outside this range, the values of $\Pr(b_i|\theta_i)$ were given by the expert elicited distributions at $\theta_i = 0$ and $\theta_i = 1$, respectively, i.e. for $\theta_i \in (0, \eta), \Pr(b_i|\theta_i) = \Pr(b_i|\theta_i = 0)$ and for $\theta_i \in (\mu, 1), \Pr(b_i|\theta_i) = \Pr(b_i|\theta_i = 1)$.

For the questions about persistence of the species group with and without management, we used existing information to provide initial values. Species were assigned a preliminary probability of persistence under no management using expert-derived estimates from the SoS database. Persistence if the KTP is removed was estimated using the estimates of relative threat importance contained in an expert-derived OEH dataset which scores threats to each species as either high, medium or low. Where a threat was not included in the dataset, it was given a medium score. These scores were weighted using the threat impact scores contained in the IUCN Threat Classification Scheme (V3.2) (http://www.iucnredlist.org/technical-documents/classification-schemes/threats-classification-scheme). The weighted scores were used to calculate the relative threat importance by dividing the weight of the threat by the sum of threat weights across all threats affecting the species. The expected persistence if the threat was removed was computed using the formula: $\Pr(\text{threat managed}) = \Pr(\text{unmanaged}) + (0.95 - \Pr(\text{unmanaged}))*\text{relative importance of threat}$. This approach assumes that if all threats were removed then persistence would be 95% (consistent with the SoS project used to calculate the persistence with no management). In making this estimate we have assumed that management completely removes the threat and we made the simplifying assumption of ignoring the influence of interactions between threats. We made this assumption because it was not feasible for experts to parameterise joint distributions for all combinations of threat interactions during an elicitation. Finally, the expected persistence probabilities for each species in the species group were averaged to obtain an overall expected average persistence for the species group. Experts had the option to change these guidance estimates if they did not agree with the predicted value.

**Computing EV_uncertainty and EVPI**. In the sections below, we firstly describe the general case for calculating $\text{EV}_\text{uncertainty}$ and EVPI for any two KTPs. Because there are many possible combinations of two KTPs, we instead computed $\text{EV}_\text{uncertainty}$ and EVPI for each KTP compared to doing nothing; this calculation is described in the remaining sections.

**General case: two Key Threatening Processes**. Here we consider the situation where managers have two possible actions: manage KTP1 or manage KTP2. This decision is trivial if the species in a species group are only affected by one of the KTPs, but for species groups affected by both KTPs, the decision requires evaluating the EVPI. We first describe the utility of the best choice under uncertainty, which corresponds to the term $\text{EV}_\text{uncertainty}$ in Eq. (1).

To simplify the computation, we assumed that the persistence when no management action is applied is perfectly known. In practice we fixed this value to $E(b_{0i}) = \overline{b_{0i}}$. In this case, the benefit was given by $E(b_i - \overline{b_{0i}})$, where $E(.)$ denotes the expectation over the distribution of $b_i$. Because $b_{0i}$ follows a beta distribution with parameters $(\alpha_0, \beta_0)$, the value of $E(b_{0i})$ can be computed analytically, i.e. $E(b_0) = \frac{\alpha_0}{\alpha_0 + \beta_0}$.

Under current uncertainty, managers do not know the outcome of actions in advance, so would choose the action which is most likely to lead to the greatest expected gain in persistence, i.e. the action which maximises:

$$\text{EV}_\text{uncertainty}(\text{KTP1}, \text{KTP2}) = \max\left(E(b_1 - \overline{b_{01}}), E(b_2 - \overline{b_{02}})\right) \quad (4)$$

The expected value is defined by $E(f(x)) = \int f(x)\Pr(x)\mathrm{d}x$, and we have that $\Pr(b_1|\overline{b_{0i}}) = \int_{\theta_1} \Pr(b_1|\theta_1, \overline{b_{0i}})P(\theta_1)\mathrm{d}\theta_1$, so we obtain:

$$\text{EV}_\text{uncertainty}(\text{KTP1}, \text{KTP2}) = \max\Bigg(\int_{\theta_1}\int_{b_1} (b_1 - \overline{b_{01}}) \Pr(b_1|\theta_1, \overline{b_{01}}) \Pr(\theta_1)\mathrm{d}\theta_1\mathrm{d}b_1,$$
$$\int_{\theta_2}\int_{b_2} (b_2 - \overline{b_{02}}) \Pr(b_2|\theta_2, \overline{b_{02}}) \Pr(\theta_2)\mathrm{d}\theta_2\mathrm{d}b_2\Bigg)$$
$$(5)$$

In practice, we assumed that $b_i$ is independent of $\overline{b_{0i}}$ and used the expert estimates of persistence with and without management to compute $\Pr(b_1|\theta_1)$.

Although the expected benefit in Eq. (5) looks complex, it has a relatively simple explanation. The integrals have the effect of doing an exhaustive search of all possible outcomes from taking management action and doing nothing. For each of the possible outcomes, the equation evaluates the gain in benefit $(b_i - \overline{b_{0i}})$ and multiplies it by the probability that that outcome will occur. Finally, all the outcomes are summed to give a total expected benefit from taking action. Equation (5) computes the largest gain in persistence we can expect from implementing either of two KTP management actions, given uncertainty about how effective KTP management actions may be at reducing the threat and the response of the species given this reduction in threat. These values can be compared across KTPs and species groups to decide the relative value of managing particular KTPs or species

groups. This value also provides a baseline that we can use to determine how much removing uncertainty would improve our management outcomes.

Next, we consider the case where uncertainty is eliminated, i.e. decision makers know in advance how effective their management will be ($\theta_i$) and the persistence they will receive from management ($b_i$). In this case, managers can choose between managing KTP1 or KTP2 because they know which one leads to the larger $b_i - \overline{b_{0i}}$. $EV_{certainty}$ provides the expectation of removing uncertainty over all possible combinations of $b_1, b_2, \theta_1, \theta_2$. We can express this as:

$$\mathrm{EV}_{\mathrm{certainty}}(\mathrm{KTP1}, \mathrm{KTP2}) = E\big(\max\big(b_1 - \overline{b_{01}}, b_2 - \overline{b_{02}}\big)\big) \quad (6)$$

Assuming again that the $b_i$ are independent of the $\overline{b_{0i}}$, we obtain:

$$\mathrm{EV}_{\mathrm{certainty}} = \int_{\theta_2}\int_{b_2}\int_{\theta_1}\int_{b_1} \max(b_1 - \overline{b_{01}}, b_2 - \overline{b_{02}})\Pr(b_1|\theta_1)\Pr(\theta_1)\Pr(b_2|\theta_2)\Pr(\theta_2)db_1 d\theta_1 db_2 d\theta_2$$
$$(7)$$

The EVPI can then be calculated from Eqs. (5) and (7), i.e.:

$$\mathrm{EVPI} = \mathrm{EV}_{\mathrm{certainty}} - \mathrm{EV}_{\mathrm{uncertainty}} \quad (8)$$

That is, the EVPI is the expected additional gain in persistence (relative to the benefits of managing under current uncertainty) that would be achieved if uncertainty about management efficiency and species response to threat reduction was removed.

**Comparing the value of action to doing nothing.** The general case described above can be used to compute the EVPI for two KTPs or species groups but pairwise comparisons are difficult to summarise (with 20 KTPs, there are 190 pairwise comparisons, excluding comparing KTPs with themselves; the number of comparisons is much larger for species groups). For simplicity and ease of computation, we instead computed the EVPI of each species group and KTP by considering that the manager has the choice between two actions: manage the KTP or do nothing. This approach provides a value of uncertainty reduction for each species group KTP: the species group or KTP for which reducing uncertainty is the most profitable will be the one with the largest value of uncertainty reduction. At the KTP level, this requires only 20 EVPI evaluations to compare all KTPs. And since in this case, the EVPI expression is simpler, we are also able to relax the assumption that doing nothing always results in the mean persistence for a species, and instead allow $b_0$ to be stochastic, allowing us to consider the effect of the counterfactual (i.e. what would have happened in the absence of management[35]) in our calculations. The remainder of the document, including the EVPXI development below and the results in the manuscript, consider this special case.

Under existing uncertainty, managers can choose to apply best practice management or to do nothing for each species group. If managers do nothing, then the benefit, or gain in persistence, will be 0. If they apply best practice management, then the expected benefit is given by $E(b_1 - b_0)$. Managers must choose between accepting a gain of 0 from doing nothing or the average gain they can expect from management, calculated over all possible outcomes of $b_0$ and $b_1$. Under current uncertainty, managers do not know the outcome of actions in advance, so would choose the action which is most likely to lead to the greatest expected gain in persistence, i.e. the action which maximises:

$$\mathrm{EV}_{\mathrm{uncertainty}} = \max(0, E(b_1 - b_0)) \quad (9)$$

Since the expectation has the property that $E(x - y) = E(x) - E(y)$, we can expand the terms in Eq. (9) to obtain an expression for the expected benefit under uncertainty:

$$\mathrm{EV}_{\mathrm{uncertainty}} = \max\left(0, \int_{b_1}\int_{\theta_1} b_1\Pr(b_1|\theta_1)\Pr(\theta_1)d\theta_1 db_1 - E(b_0)\right) \quad (10)$$

Because $b_0$ follows a beta distribution with parameters $(\alpha_0, \beta_0)$, the value of $E(b_0)$ can be computed analytically, i.e. $E(b_0) = \frac{\alpha_0}{\alpha_0 + \beta_0}$.

Next we consider the case where uncertainty is eliminated, i.e. decision makers know in advance how effective management will be and the gains in persistence they will receive from management. In this case the decision-maker can choose the best (i.e. max) decision for an outcome before we multiply by the probability of that outcome. We can express this as

$$\mathrm{EV}_{\mathrm{certainty}} = E(\max(0, b_1 - b_0)) \quad (11)$$

Applying the definition of the expectation and using Eq. (3), we obtain:

$$\mathrm{EV}_{\mathrm{certainty}} = \int_{b_1}\int_{\theta_1}\int_{b_0} \max(0, b_1 - b_0)\Pr(b_1|\theta_1, b_0)\Pr(\theta_1)\Pr(b_0)db_0 d\theta_1 db_1 \quad (12)$$

Because the integrand is zero whenever $b_1 < b_0$, we can simplify the computation of Eq. (12) by reducing the range of the integral over $b_1$:

$$\mathrm{EV}_{\mathrm{certainty}} = \int_{b_0=0}^{1}\int_{\theta_1=0}^{1}\int_{b_1=b_0}^{1} (b_1 - b_0)\Pr(b_1|\theta_1, b_0)\Pr(\theta_1)\Pr(b_0)db_1 d\theta_1 db_0 \quad (13)$$

Finally, the EVPI can be calculated from Eq. (8), assuming independence between $b_1$ and $b_0$ as for Eq. (7).

**Computing the EVPXI.** The EVPXI describes the expected gain in persistence if some components of the uncertainty are resolved but the other components remain uncertain. In this section we derive separate formulae for EVPXI for the cases where managers must choose between managing a KTP or doing nothing and are certain about: (i) the effectiveness of management to remove threats ($\theta$-certainty); (ii) the response of species to management ($b_1$-certainty); or (iii) the response of species in the absence of management ($b_0$-certainty). In all cases $EV_{uncertainty}$ remains the same as for EVPI (Eq. 10). The logic for obtaining the EVPXI expressions is similar to that used to derive $EV_{certainty}$.

*Case 1: EVPXI with $\theta$-certainty:* In this case, we have resolved the uncertainty about the effectiveness of managing the threat, but are uncertain about the species response. We can write the expectation for $\theta$-certainty as

$$\mathrm{EV}_{\theta-\mathrm{certainty}} = E_\theta\Big(\max\big(0, E_{b_1, b_0|\theta}(b_1 - b_0)\big)\Big) \quad (14)$$

where

$$E_{b_1, b_0|\theta}(b_1 - b_0) = \int_{b_1}\int_{b_0}(b_1 - b_0)\Pr(b_1|\theta_1)\Pr(b_0)db_0 db_1 \quad (15)$$

Let $I\big(E_{b_1, b_0|\theta}\big) = \begin{cases} 1, & \text{if } E_{b_1, b_0|\theta} > 0 \\ 0, & \text{otherwise} \end{cases}$

Then

$$\max\big(0, E_{b_1, b_0|\theta}(b_1 - b_0)\big) = I\big(E_{b_1, b_0|\theta}\big)\left(\int_{b_1}\int_{b_0}(b_1 - b_0)\Pr(b_1|\theta_1)\Pr(b_0)db_0 db_1\right) \quad (16)$$

So

$$\mathrm{EV}_{\theta-\mathrm{certainty}} = \int_\theta\left(\int_{b_0}\int_{b_1}(b_1 - b_0)\Pr(b_1|\theta_1)\Pr(b_0)db_1 db_0\right) I\big(E_{b_1, b_0|\theta}\big)\Pr(\theta)d\theta \quad (17)$$

We can now compute the EVPXI using:

$$\mathrm{EVPXI}_{\theta-\mathrm{certainty}} = \mathrm{EV}_{\theta-\mathrm{certainty}} - \mathrm{EV}_{\mathrm{uncertainty}} \quad (18)$$

*Case 2: EVPXI with $b_1$-certainty:* In this case, we have resolved the uncertainty about the response of species to management, but remain uncertain about the effectiveness of threat reduction and about the response of species in the absence of management. We can write the expectation for $b_1$-certainty as:

$$\mathrm{EV}_{b1-\mathrm{certainty}} = E_{b_1}\Big(\max\big(0, E_{\theta, b_0|b_1}(b_1 - b_0)\big)\Big) \quad (19)$$

where

$$E_{\theta, b_0|b_1}(b_1 - b_0) = \int_{\theta_1}\int_{b_0}(b_1 - b_0)\Pr(\theta_1)\Pr(b_0)db_0 d\theta_1 \quad (20)$$

Let $I\big(E_{\theta, b_0|b_1}\big) = \begin{cases} 1, & \text{if } E_{\theta, b_0|b_1} > 0 \\ 0, & \text{otherwise} \end{cases}$

Then

$$\begin{aligned}
\max\big(0, E_{\theta, b_0|b_1}(b_1 - b_0)\big) &= I\big(E_{\theta, b_0|b_1}\big)\int_{\theta_1}\int_{b_0}(b_1 - b_0)\Pr(\theta_1)\Pr(b_0)db_0 d\theta_1 \\
&= I\big(E_{\theta, b_0|b_1}\big)\int_{b_0}(b_1 - b_0)\Pr(b_0)db_0 \\
&= I\big(E_{\theta, b_0|b_1}\big)(b_1 - E(b_0)) \\
&= I(b_1 > E(b_0))(b_1 - E(b_0))
\end{aligned} \quad (21)$$

So

$$\begin{aligned}
\mathrm{EV}_{b_1-\mathrm{certainty}} &= \int_{b_1}(b_1 - E(b_0))I\big(E_{\theta, b_0|b_1}\big)\Pr(b_1)db_1 \\
&= \int_{\theta_1}\int_{b_1}\Big(I\big(E_{\theta, b_0|b_1}\big)(b_1 - E(b_0))\Big)\Pr(b_1|\theta_1)\Pr(\theta_1)db_1 d\theta_1
\end{aligned} \quad (22)$$

We can now compute the EVPXI using:

$$\mathrm{EVPXI}_{b1-\mathrm{certainty}} = \mathrm{EV}_{b1-\mathrm{certainty}} - \mathrm{EV}_{\mathrm{uncertainty}} \quad (23)$$

Case 3: EVPXI with $b_0$-certainty: In this case, we have resolved the uncertainty about the response of species in the absence of management, but remain uncertain about threat reduction effectiveness and the response of species to management. We can write the expectation for $b_0$-certainty as:

$$\mathrm{EV}_{b0-\mathrm{certainty}} = E_{b_0}\Big(\max\big(0, E_{\theta, b_1|b_0}(b_1 - b_0)\big)\Big) \quad (24)$$

where

$$E_{\theta, b_1 | b_0}(b_1 - b_0) = \int_{\theta_1} \int_{b_1} (b_1 - b_0) \Pr(\theta_1)\Pr(b_1|\theta_1) db_1 d\theta_1 \tag{25}$$

Let $I\left(E_{\theta, b_1|b_0}\right) = \begin{cases} 1, & \text{if } E_{\theta, b_1|b_0} > 0 \\ 0, & \text{otherwise} \end{cases}$

Then

$$\max\left(0, E_{\theta, b_1|b_0}(b_1 - b_0)\right) = I\left(E_{\theta, b_1|b_0}\right) \int_{\theta_1} \int_{b_1} (b_1 - b_0)\Pr(\theta_1)\Pr(b_1|\theta_1)\,db_1 d\theta_1$$

$$= I\left(E_{\theta, b_1|b_0}\right)\left(\int_{\theta_1}\int_{b_1} b_1 \Pr(\theta_1)\Pr(b_1|\theta_1)db_1 d\theta_1 - b_0\right) \tag{26}$$

So

$$EV_{b_1-\text{certainty}} = \int_{b_0} I\left(E_{\theta, b_1|b_0}\right)\left(\int_{\theta_1}\int_{b_1} b_1\Pr(\theta_1)\Pr(b_1|\theta_1)db_1 d\theta_1 - b_0\right)\Pr(b_0)db_0 \tag{27}$$

We can now compute the EVPXI using:

$$EVPXI_{b0-\text{certainty}} = EV_{b0-\text{certainty}} - EV_{\text{uncertainty}} \tag{28}$$

**Ethics statement**. Ethics approval was obtained from the CSIRO Social Science Human Research Ethics Committee prior to conducting elicitation (project ID: 006/18).

**Reporting summary**. Further information on research design is available in the Nature Research Reporting Summary linked to this article.

## Data availability

All data for this manuscript, including anonymised expert inputs and R scripts, is available from Figshare: https://doi.org/10.6084/m9.figshare.7623665. The source data underlying Figs. 1–2, 4–7 and Supplementary Figs 1–4 are provided as a Source Data file.

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

## Acknowledgements

The authors would like to thank the species experts who contributed their considerable expertise and valuable time to this project. We also thank Hannah Lloyd (Office of Environment and Heritage) for discussions and support throughout the project.

## Author contributions

I.C., J.B.-B., S.N., E.G. and A.M. conceived and designed the study. J.B.-B., A.M., E.G. provided initial species groupings and collated species information. I.C. and S.N. conducted the elicitation. S.N., I.C. and N.P. developed the VOI formulations. S.N. conducted the analysis. S.N. led the writing of the draft manuscript; all authors provided edits to generate the final manuscript.

## Additional information

**Competing interests:** This work was co-funded by the Office of Environment and Heritage's Saving our Species programme and CSIRO (SN, IC). The remaining authors declare no competing interests.

