## [Peer Review File · Nature Communications]

Reviewers' Comments:

Reviewer #1:

Remarks to the Author:

In this paper, the authors describe an analysis of the relative importance of reducing uncertainties in the context of managing 976 species and ecological communities in New South Wales, Australia. Building on several recent papers on threats-based prioritization for conservation management, this paper shows how value-of-information calculations can be used to identify the uncertainties that most impede such decisions. The methods are applied to a case study of impressive scope, complexity, and importance.

The paper is lucid, concise, and compelling. The figures are informative and well constructed. I appreciate how much care the authors took in preparing the text, which will require very little attention from the copyeditors. The Supplementary Information is extensive and detailed and provides clear technical documentation of the methods.

Many existing analyses of value-of-information in conservation settings have found the value of perfect information to be surprising small. By taking on such a comprehensive analysis across many species, the authors show that the cumulative value of reducing uncertainty can be quite substantial. This is an important perspective.

I have only one concern, but it will require a response in several parts of the paper. The key methods are described in Section 6.2 of the Supplementary Material; I was not able to understand how the analysis was undertaken until I had read that section. Two suggestions arise from this:

First, Section 4 in the main body needs to explain more clearly the structure of the decisions that were analyzed, namely, that the expected benefit and the value of information concern the binary decision whether or not to implement an action to mitigate a particular KTP. This is mentioned briefly at lines 194-195, but I couldn't understand it until I had read the supplemental information.

Second, the limitations of this approach need to be discussed in more detail. The inference the authors want to make is a comparison across KTPs (as an extension of the methods describe in Section 6.1) and much of the main body reports conclusions as if that was the comparison analyzed. I suspect that the methods of 6.2 hold up for comparing the expected benefit in the face of uncertainty across KTPs, but I'm not convinced they do for comparing the EVPI across KTPs. Here's the thought experiment: suppose you have a set of KTPs, and the uncertainty in the benefit associated with addressing them is large but has nearly all of its probability density above 0. In the analysis of each individual KTP (comparing best management practice with no action, as in 6.2), the EVPI will be essentially zero, because there is a demonstrable benefit in acting. The comparison of mean benefit across KTPs is valid as a way of ranking them in the face of uncertainty. But the comparison of EVPI across KTPs suggests there is not value in resolving uncertainty. The distributions of the benefits associated with addressing each KTP, however, may very well overlap each other, even if none of them overlap zero. So, there could be value in resolving uncertainty about which KTP is more important to address, even though there is not doubt that acting is better than not acting. The point is: the decision structure the authors analyzed is not exactly the decision structure they are inclined to make an inference about, so the conclusions have to be handled delicately. Unless I misunderstood something, which is possible.

I appreciate the opportunity to have read this paper and commend the authors on their thoughtful and excellent work.

Michael C. Runge, U.S. Geological Survey, Patuxent Wildlife Research Center

Reviewer #2:

Remarks to the Author:

The application of VOI to biodiversity management has the potential to improve the rate of return on management interventions. While the paper is carefully done, there are some important limitations that lead me to reject the paper.

(1) The external validity of the study is not immediately clear given the experts and the focal system of the study. The method is applicable to all settings but the conclusions will have limited use outside of their system.

(2) The use of expert elicitation is a serious weakness and raises important concerns on the internal validity of the study. One issue is the low response rate of the experts (66 out of 250), which raises questions about sample selection (What are the characteristics of those who replied? Do these characteristics correlate with the nature of their responses?). Another issue is the lack of calibration in the expert elicitation, see Colson and Cooke (citation below) for a review article on validation of expert elicitation. Finally, the small samples in table 3 in the SI, raises questions on the informational value of the content used in this exercise.

(3) Other serious limitations of the study are: (a) the independence across management interventions; (b) the assumption that management can do no harm; (c) the description of a zero-one for management intervention (nothing or all); and (4) complete lack of information on the cost of the different management interventions (we are interested in the net rate of return from the investment in management not just the VOI).

(4) While not covered in the analysis, it could very well be the most important piece of uncertainty to resolve is on the cost of the management interventions or on the threat interactions. Unfortunately, these factors were abstracted away from in the current expert solicitation, and so there is no guidance on their relative importance.

(5) Finally, line 287, concludes that "implementing management without understanding the species of interest is limiting conservation effectiveness." Do we really need all of this machinery to make this conclusion? Seems blatantly obvious to anyone who spends anytime thinking about species management.

While I think there is a lot of improvement to be made in species management and I think VOI can have a role to play in that improvement, I think the issues of external and internal validity of this study are too great to recommend publication.

Citation:

Abigail R Colson, Roger M Cooke; Expert Elicitation: Using the Classical Model to Validate Experts' Judgments, *Review of Environmental Economics and Policy*, Volume 12, Issue 1, 1 February 2018, Pages 113–132, <https://doi.org/10.1093/reep/rex022>

Reviewer #3:

Remarks to the Author:

This study investigates the importance of uncertainty when managing threats to species in New South

Wales, Australia, using expert elicitation and Value of Information calculations. The authors were looking to identify whether management action would change if current uncertainty around threats to species or the effectiveness of management were reduced. They found that some threats were well-understood, and further information about threats would add little in terms of management gain, whilst others were poorly understood and therefore would benefit from further research. This is the first study that I am aware of that aims to identify priority actions using Value of Information for such a broad range of species. Value of Information has great potential to be of use to conservation scientists but to date has mainly been used for studies on single-species management. The authors have shown how Value of Information could be applied much more broadly as a way to identify research and management priorities for many threatened species at once which would be of interest not only to conservation scientists, but also to policy makers.

The methods are mainly described in sufficient detail and are appropriate. The authors undertook quite a number of calculations which are well explained in the appendix. It would be useful to explain how data were collected first before explaining the calculations in the supplementary material, so that the reader gets an understanding of the data first. Could you also add the number of species in table 3 in the appendix?

The results presented in sections 3.2 to 3.5 are interesting. It would be useful if the key findings would be presented more succinctly, and possibly also add some case studies about the species and their threats, as you did in section 3.6, lines 233 – 240. I appreciate your discussion of the interaction between threats and can see why an analysis of all interactions would be challenging. I also like figure 6, it shows the interactions well. It might be useful to just compare two threats, as a case study, to show how the calculations could be done. It would make sense to compare fire with another threat for example, given how prevalent fire seems to be in the interactions.

Figures 1 - 5 present the overall values from the expert elicitation, but in the main text the authors talk about percentages, so some clarification on what these percentages are would be useful. To this end, it might help to change the figures to those percentages if possible. Figures 1 – 5 also showed both the sum of values as well as per-species values, but then there is little discussion around why both were included, so I suggest to either simplify the figures and only show the sums, or add more discussion around why the per-species values were included. Is figure 3 showing some of the information also shown in figure 1? This was not quite clear to me.

Some abbreviations or terms were not explained in the main text or in the supplementary material – what does TEC stand for? Please add what this means to the figure captions too. What are critical weight mammals? Could KTP be changed to threats?

I found the section on EVPXI difficult to follow because of how the information is presented in figure 5. It would help if figure 5 would show the 3 bars of the uncertainties next to each other. Where is the green bar shown in the legend?

Fire seems to stand out from the other threats in that there is high uncertainty around how it can be managed. However, it seems that this is based on the expert judgement by one person only. This would be worth mentioning in the discussion.

This is a well-written manuscript, particularly the introduction and section 3.6 are to the point and cover what is most important, with appropriate use of references. The authors make a good case for why this is important work. The abstract is clear and concise.

Reviewers' comments:

Reviewer #1 (Remarks to the Author):

In this paper, the authors describe an analysis of the relative importance of reducing uncertainties in the context of managing 976 species and ecological communities in New South Wales, Australia. Building on several recent papers on threats-based prioritization for conservation management, this paper shows how value-of-information calculations can be used to identify the uncertainties that most impede such decisions. The methods are applied to a case study of impressive scope, complexity, and importance.

The paper is lucid, concise, and compelling. The figures are informative and well constructed. I appreciate how much care the authors took in preparing the text, which will require very little attention from the copyeditors. The Supplementary Information is extensive and detailed and provides clear technical documentation of the methods.

Many existing analyses of value-of-information in conservation settings have found the value of perfect information to be surprising small. By taking on such a comprehensive analysis across many species, the authors show that the cumulative value of reducing uncertainty can be quite substantial. This is an important perspective.

Response: We thank the reviewer for their positive comments about our work.

I have only one concern, but it will require a response in several parts of the paper. The key methods are described in Section 6.2 of the Supplementary Material; I was not able to understand how the analysis was undertaken until I had read that section. Two suggestions arise from this:

Response: Agreed. We have addressed the two specific suggestions below and have also attempted to add clarifications earlier in the text. To further improve the clarity, we have added the definitions of $EV_{\text{uncertainty}}$ and $EV_{\text{certainty}}$ from SI section 6.2 to section 4 (lines 369-371); this will remove ambiguity for readers interested in the method without requiring the full set of equations in SI section 6.2.

1) First, Section 4 in the main body needs to explain more clearly the structure of the decisions that were analyzed, namely, that the expected benefit and the value of information concern the binary decision whether or not to implement an action to mitigate a particular KTP. This is mentioned briefly at lines 194-195, but I couldn't understand it until I had read

the supplemental information.

Response: Accepted. We have added a sentence to the start of the methods section 4 that states: “The decision problem we considered was to choose whether to manage a given threat or not. This required evaluating the expected gain in persistence from managing a threat, compared to doing nothing, when the outcomes of both management and doing nothing were uncertain.” (lines 351-353). We have also added a reference to SI section 6.2 in the methods section and a sentence to lines 49-51 to ensure that the decision problem is made clear to readers early in the manuscript, i.e.: “The decision problem we analyse is whether or not to manage a KTP, given uncertainties about the effectiveness of management and about what would happen to species groups without management.”

2) Second, the limitations of this approach need to be discussed in more detail. The inference the authors want to make is a comparison across KTPs (as an extension of the methods describe in Section 6.1) and much of the main body reports conclusions as if that was the comparison analyzed. I suspect that the methods of 6.2 hold up for comparing the expected benefit in the face of uncertainty across KTPs, but I'm not convinced they do for comparing the EVPI across KTPs. Here's the thought experiment: suppose you have a set of KTPs, and the uncertainty in the benefit associated with addressing them is large but has nearly all of its probability density above 0. In the analysis of each individual KTP (comparing best management practice with no action, as in 6.2), the EVPI will be essentially zero, because there is a demonstrable benefit in acting. The comparison of mean benefit across KTPs is valid as a way of ranking them in the face of uncertainty. But the comparison of EVPI across KTPs suggests there is not value in resolving uncertainty. The distributions of the benefits associated with addressing each KTP, however, may very well overlap each other, even if none of them overlap zero. So, there could be value in resolving uncertainty about which KTP is more important to address, even though there is no doubt that acting is better than not acting. The point is: the decision structure the authors analyzed is not exactly the decision structure they are inclined to make an inference about, so the conclusions have to be handled delicately. Unless I misunderstood something, which is possible.

Response: Agreed—this is true and we thank the reviewer for pointing this out. As we argue below, there are two different decision problems at play here, and each of them has its own associated definition of the ‘value’ of information. As we will explain, our work deals only with one of them. But, firstly, to help visualise the thought problem posed by the reviewer, we created the illustration below, which also shows the difference between the two decision problems.

There are two decision problems being discussed here, with different ‘values’ being computed in the value of information calculation because they answer different questions. The first problem (which we solved in our paper) is a decision about whether or not to manage. Here the uncertainty is around what would have happened with management (uncertainty about management effectiveness) and without management (uncertainty on persistence without management, i.e. the counterfactual); for this decision problem, resolving these uncertainties determines the EVPI. An EVPI of 0 means that we are certain that managing is better than doing nothing; positive EVPI values mean that there is some value in resolving the uncertainty about management effectiveness and/or the counterfactual. The EVPI is calculated relative to a common benchmark for all KTPs (i.e. compared to doing no management) and is comparable across KTPs in the sense that it computes the expected absolute additional gain in persistence that could be obtained by resolving uncertainty about management effectiveness. It is valid to compare the EVPI results relative to this common benchmark: they tell us how much we would gain from resolving the uncertainties for each KTP when considering the two options management and doing nothing. As they have a common benchmark, we can rank the KTPs as in figures 1, 2, 4 and 5. However, we agree with the reviewer that we cannot determine the relative value of managing one KTP compared to another (we were careful to avoid these comparisons in our manuscript).

The second decision problem, raised by the reviewer, is a decision about *which* KTP to manage, conditional on the premise that we will manage one of the KTPs. Using the reviewer’s example, if we have two KTPs that both have EVPI=0 in the first decision problem, then we know that resolving the uncertainty about management effectiveness is not valuable—because the counterfactual does not impede our decision and managing is better than doing nothing. However, if we had already decided to manage one of the KTPs (in this case, we know that this is a good idea, since we know for sure that management is better than doing nothing), then there would be value in deciding which to manage (i.e. the second decision problem). This is a relative EVPI, which is computed on a pairwise basis for any particular combination of KTPs: i.e. what is the benefit of resolving uncertainty about management effectiveness between two KTPs? We derived the formulation for the 2-KTP

problem in section 6.1 of the supporting information, but did not present results because computing the EVPI for every pairwise combination of KTPs would be both extremely computationally expensive and extremely difficult to present succinctly (there are 190 pairwise comparisons for KTPs and many more for species groups; see text at lines 142-144 of the SI and lines 311-316 of the SI). It is also worth noting that the formulation of section 6.1 only considers pairwise comparisons—the EVPI results would change again (and the number of required comparisons would increase exponentially) if we considered a decision set including 3 or more KTPs. As a result of these complications, we suggest that this decision problem is best solved on a case-by-case basis when comparing two specific KTPs rather than being solved globally (lines 258-265 of the main text of the revision).

To summarise—the reviewer raises a different decision problem to the one we solved in our paper. We are confident that our findings and comparisons in the paper remain valid, but agree that the problem suggested by the reviewer is an important distinction that is worth clarifying. To address this we have added a paragraph to the caveats in the discussion which summarises the arguments above and what they mean for the interpretation of our results (lines 258-265).

I appreciate the opportunity to have read this paper and commend the authors on their thoughtful and excellent work.

Michael C. Runge, U.S. Geological Survey, Patuxent Wildlife Research Center

Response: We thank the reviewer again for their informed review and for supporting our work.

Reviewer #2 (Remarks to the Author):

The application of VOI to biodiversity management has the potential to improve the rate of return on management interventions. While the paper is carefully done, there are some important limitations that lead me to reject the paper.

(1) The external validity of the study is not immediately clear given the experts and the focal system of the study. The method is applicable to all settings but the conclusions will have limited use outside of their system.

Response: Clarification. We agree that our method is a key selling point for our study and is applicable to all settings. However we disagree that the selected experts or the case study limit the importance of our work. As outlined in the conclusion (lines 322-349), we have three major findings that are important regardless of our experts or our case study:

- 1) Our study is the first to quantify the substantial impediment that uncertainty poses to effective management of key threatening processes. No other study has tackled such a wide range of threats, nor found the value of information to be as significant as in our study. Although conservation biologists intuitively know that uncertainty impedes the effectiveness of our management, to date there have been no value of information studies that adequately quantify this. By undertaking our analysis at a large scale and incorporating so many species, we were able to demonstrate for the first time that uncertainty does indeed play a critical role in management effectiveness. Importantly,

we were also able to quantify the extent of the impact of uncertainty, which has not previously been demonstrated. Although the specific numbers may change in another case study (this is true of any case study, with traditional data or experts), our study is nonetheless the first to demonstrate just how much impact uncertainty is having on management effectiveness.

- 2) We separated the different types of uncertainty and showed for the first time that uncertainty about species response was the most beneficial type of uncertainty to reduce. This will help to target efforts to improve future threatened species management actions, and is particularly relevant considering the recognised paucity of data about the responses of species to management (Sutherland *et al.* 2004). As with the first finding, this is a significant advance that quantifies the relative importance of types of uncertainty; this result was not previously known and it has not been quantified at this scale.
- 3) We found that invasive plant control and island eradications are well understood while fire management and invasive animal control would benefit from additional research. Although we acknowledge the potential for regional variation or variation in expert values, the generalised findings that are specific to our case study corroborate and support existing research by providing another line of evidence via our novel value of information approach. These findings add weight to existing (cited) literature that also found major uncertainties in these areas (Cahill *et al.* 2008; Walsh *et al.* 2012; Lazenby, Mooney & Dickman 2015; Doherty & Ritchie 2017), while also providing a method to quantify the relative importance of these findings for any location.

In the discussion, we added a paragraph that addresses the use of local experts, acknowledges that threats may be ranked differently in other parts of the world, and describes the external relevance of this study (lines 304-321).

(2) The use of expert elicitation is a serious weakness and raises important concerns on the internal validity of the study. One issue is the low response rate of the experts (66 out of 250), which raises questions about sample selection (What are the characteristics of those who replied? Do these characteristics correlate with the nature of their responses?). Another issue is the lack of calibration in the expert elicitation, see Colson and Cooke (citation below) for a review article on validation of expert elicitation. Finally, the small samples in table 3 in the SI, raises questions on the informational value of the content used in this exercise.

Response: Experts were selected based on recommendations from the New South Wales Office of Environment, and were ecologists or threatened species managers chosen based on their expertise with either KTPs or affected species groups (lines 198-200 of the SI; expertise added to the SI in response to reviewer question about expert characteristics). While there are low numbers of respondents for some of the KTPs, this is reflective of the number of people involved in managing and making decisions about management of threatened species—there are often only a handful of scientists and managers who are responsible for implementing and studying a particular threat and species response. Notwithstanding this, we do acknowledge that some of the KTPs have low numbers of respondents, and therefore our findings may be sensitive to change if repeated with different experts, particularly in different jurisdictions where threats may have different relative importance (although note our response to the previous comment, which argues that our study still makes a number of findings that are important regardless of the experts chosen). We have added two paragraphs to the discussion that discuss this issue in more detail (lines 304-321).

Our response rate (25%) was not particularly low for an email survey. In a meta-analysis of email vs traditional survey methods, Shih and Fan (2009) found that the average response rate for email surveys was 33% (STD: 22%, N=35 studies). Our elicitation exercise was difficult and time-consuming relative to a standard survey, and no incentives were offered to complete the work, so we expected a lower response rate than for a more conventional questionnaire-type survey.

We sought to address calibration by providing experts with precomputed persistence estimates based on the number and severity of threats (see lines 54-69/Figure 2 of the supporting information). In the absence of data required for 'true' calibration questions, we used this approach to ensure that experts had an estimate that summarised our best knowledge of the system when making their estimates. As these precomputed estimates were based on quite broad assumptions, we had no basis to assume that they were more or less valid than an experienced expert prediction and we did not consider that these precomputed estimates were suitable for calibration in the manner of the classical model of Colson and Cooke. The remainder of our elicitation procedure was designed to minimise expert judgement biases, following a well-cited approach that has been demonstrated to work well in similar elicitation situations to our study and when limited data are available (Speirs-Bridge *et al.* 2010).

(3) Other serious limitations of the study are: (a) the independence across management interventions; (b) the assumption that management can do no harm; (c) the description of a zero-one for management intervention (nothing or all); and (d) complete lack of information on the cost of the different management interventions (we are interested in the net rate of return from the investment in management not just the VOI).

Response: We acknowledge (a); this was a limiting assumption of our work and is discussed in detail in lines 276-303. We have added additional detail about why we made the assumption of independent management of threats in response to a request from reviewer 3 (lines 278-281). However, while we acknowledge this assumption, there is no feasible way to overcome the assumption of independence of threats, because the way that threats interact is highly uncertain and extremely difficult to quantify (see also response to reviewer 3 regarding interactions). In the absence of information about interacting threats, our method provides a significant improvement on the current state of knowledge and provides a useful way to prioritise among threats using the information available.

We disagree that (b) is a serious limitation of this study. Our method does not preclude negative benefits (i.e. harmful actions that result in lower benefits than doing nothing) and we did not make this assumption in our approach. Our experts generally estimated that taking action has a higher mean benefit than doing nothing. This does not seem contentious, since the management of threatened species is unlikely to be more harmful than not taking action, particularly in the scenario of threatened species, where the alternative (i.e. doing nothing) may condemn the species to extinction.

Point (c) can be interpreted two ways. Firstly, it may be intended as a criticism of the comparison of a single action to doing nothing, which we have addressed in our response to reviewer 1 above. We provided a general method to compare the value of information with different benchmarks (e.g. compare two different management actions), but as explained in section 6.1 of the Supporting Information (and referenced at lines 212-215 of the text), the number of potential combinations makes this procedure impractical to summarise into digestible findings. Our approach is a valid way to compare across different threats with a common benchmark.

Secondly, the reviewer could be alluding to a need to model partial management effectiveness. If so, then our method accommodates this already. We allowed for various degrees of management effectiveness via the level of management effectiveness θ_i reported

as a proportion of the initial threat removed after management (see Supporting Information S1: Section 4—Uncertainty Model). This parameter covers the range of possible management outcomes; each of which has a probabilistic species response. We did compare the value of information of each threat to a benchmark of doing nothing, but this was largely to ensure a common baseline so that we can compare across threats and does not mean that we ignored the possibility of partial management effectiveness.

We acknowledge (d); this is explained as a limitation of our work in lines 267-275. We elaborate on the issue of cost in the next point raised by the reviewer.

(4) While not covered in the analysis, it could very well be the most important piece of uncertainty to resolve is on the cost of the management interventions or on the threat interactions. Unfortunately, these factors were abstracted away from in the current expert solicitation, and so there is no guidance on their relative importance.

Response: The remark about threat interactions has been addressed in the response to assumption (a) above.

Costs are discussed in our manuscript at lines 267-275. We agree with the reviewer that costs are an important aspect to decision making when decision makers seek to make cost-effective decisions (this may not always be the objective with threatened species, for example where certain charismatic species (such as koalas or pandas) might be managed despite their comparatively low cost-effectiveness). Roughly speaking, cost-effective decisions for species management can be made using three pieces of information: biodiversity 'benefits', costs and likelihood of success (Bottrill *et al.* 2008; Carwardine *et al.* 2019). Cost effectiveness is measured as the product of benefit and likelihood of success divided by cost. Our study quantifies a metric of both biodiversity benefit and likelihood of success by determining the expected gain in species persistence that could be obtained by managing threats. By doing so, our study provides the most difficult part of a cost-effectiveness analysis: quantifying benefits to threatened species are often more difficult to estimate than costs of management intervention, which tend to be better understood and comparatively easy to estimate. Including cost would be useful but it would not require a methodological advance.

A second point is that providing benefits allows managers with different objectives to use our results; providing cost-effectiveness scores presumes that the objective is about maximising cost-effectiveness. Our findings provide the raw knowledge required to answer different objectives: for example managers who simply want to get the most benefit from management could use this to prioritise their management, while those seeking cost-effective returns could fairly easily divide the results for each threat by the cost of management to meet their goals.

(5) Finally, line 287, concludes that “implementing management without understanding the species of interest is limiting conservation effectiveness.” Do we really need all of this machinery to make this conclusion? Seems blatantly obvious to anyone who spends anytime thinking about species management.

Response: Accepted. We agree that this sentence is not necessary to support our paper's conclusions. We have removed the sentence.

While I think there is a lot of improvement to be made in species management and I think VOI can have a role to play in that improvement, I think the issues of external and internal

validity of this study are too great to recommend publication.

Citation:

Abigail R Colson, Roger M Cooke; *Expert Elicitation: Using the Classical Model to Validate Experts' Judgments*, *Review of Environmental Economics and Policy*, Volume 12, Issue 1, 1 February 2018, Pages 113–132, <https://doi.org/10.1093/reep/rex022>

Reviewer #3 (Remarks to the Author):

This study investigates the importance of uncertainty when managing threats to species in New South Wales, Australia, using expert elicitation and Value of Information calculations. The authors were looking to identify whether management action would change if current uncertainty around threats to species or the effectiveness of management were reduced. They found that some threats were well-understood, and further information about threats would add little in terms of management gain, whilst others were poorly understood and therefore would benefit from further research. This is the first study that I am aware of that aims to identify priority actions using Value of Information for such a broad range of species. Value of Information has great potential to be of use to conservation scientists but to date has mainly been used for studies on single-species management. The authors have shown how Value of Information could be applied much more broadly as a way to identify research and management priorities for many threatened species at once which would be of interest not only to conservation scientists, but also to policy makers.

Response: We thank the reviewer for their positive comments.

The methods are mainly described in sufficient detail and are appropriate. The authors undertook quite a number of calculations which are well explained in the appendix. It would be useful to explain how data were collected first before explaining the calculations in the supplementary material, so that the reader gets an understanding of the data first.

Response: The current structure of the SI first gives a method overview (section 2, including a general introduction to VOI calculations in sections 2.1 and 2.2), then introduces our specific problem in section 3 and the uncertainty model in section 4, before discussing the data collected in section 5 and then the problems-specific EVPI calculations in section 6. Note that the calculations of EVPI and EVPXI in the original submission (section 6) do follow the data collection section (section 5). However, we assume that the reviewer is suggesting that we move section 5 to a position before section 3.

Because our data was elicited for the purpose of answering a specific decision question, we feel that it makes sense to first introduce the decision problem being solved (section 3) and the uncertainty model that needs to be parameterised with data (section 4). Using this ordering allows the reader to understand why the specific expert questions were asked and how they fit into the decision problem. In contrast, discussing the data before properly motivating our problem creates problems for the text as terms and notation (e.g. benefit b_i , level of management effectiveness θ_i) need to be introduced before we know what the decision problem is, and the structure of the elicitation (SI lines 220-245) reads as complex and uncoordinated without prior motivation in section 4. Due to these structural issues, we have chosen not to move the data collection section. However, to address the reviewer concern, we have added a sentence to the first paragraph of the objectives section 3 which notes that the main sources of data were expert elicited with reference to existing databases (SI lines 131-133).

Could you also add the number of species in table 3 in the appendix?

Response: Agreed—we have added the number of species to Table 3 as requested.

The results presented in sections 3.2 to 3.5 are interesting. It would be useful if the key findings would be presented more succinctly, and possibly also add some case studies about the species and their threats, as you did in section 3.6, lines 233 – 240.

Response: We have included some discussion of specific threats in the species group sections 3.3. and 3.4 (see lines 135-146 and 165-182), but to some extent, the level of detail we go into is constrained by the short format of the journal, so we have focused more on broad trends rather than individual species group outcomes (note however that the rankings of all species groups is presented in the figures in Supporting Information S2). However we also interpret the reviewer's comment as a request to better communicate the results from the VOI analysis. To address this we have added a new figure to the text which graphically shows the results for a single KTP (loss of tree hollows, Figure 3). The figure shows the relative magnitudes of the EVPI and expected value under uncertainty at the different scales analysed in the manuscript (i.e. expected value for the entire KTP and also for each species group). This should help readers to better interpret the figures in sections 3.2-3.5 as requested by the reviewer.

I appreciate your discussion of the interaction between threats and can see why an analysis of all interactions would be challenging. I also like figure 6, it shows the interactions well. It might be useful to just compare two threats, as a case study, to show how the calculations could be done. It would make sense to compare fire with another threat for example, given how prevalent fire seems to be in the interactions.

Response: Although we can compare two KTPs assuming independence between them, this does not truly address the interactions between threats. Quantifying interactions requires the relative strengths of each threat under all combinations of management effectiveness (i.e. how much does the presence of threat x modify the influence/impact of threat y)? Currently we make the assumption that threats can be managed independently (i.e. that managing a threat does not influence the impacts of other threats on the target species). Relaxing the assumption of independent threats is difficult even for two interacting threats. This information is hard to elicit because it would require experts to parameterise the joint distribution for all combinations of threat interactions during an elicitation (see SI lines 261-263). In practice, for a pair of KTPs, this would require experts to specify the expected benefit of management for a 2-dimensional joint distribution $\Pr(\theta_1, \theta_2)$, where θ_i is the effectiveness of management of KTP i . This would require eliciting different distributions $\Pr(b_1 | \theta_1, \theta_2)$ representing the benefit of managing KTP1 given that KTP1 and KTP2 are managed with effectiveness θ_1 and θ_2 respectively. In practice this would be difficult for the experts to conceptualize and require an overwhelmingly large number of elicitations (i.e. one version of the current elicitation for each combination of (θ_1, θ_2) considered). Due to the large elicitation burden, we did not attempt to elicit this information. We have added clarifying text to the manuscript (lines 278-282 of the revision) to better explain the difficulties of quantifying interactions.

Figures 1 - 5 present the overall values from the expert elicitation, but in the main text the authors talk about percentages, so some clarification on what these percentages are would be useful. To this end, it might help to change the figures to those percentages if possible.

Response: We agree that the percentages are confusing due to the word ‘gain’ in the text. In fact the percentages represent absolute (not relative) gains, and since persistence is measured on a 0-1 scale, they are interchangeable with the persistence values in the figures. We used percentage values in the text as these are easier to conceptualise than decimal values, particularly if readers forget that persistence is measured on a 0-1 scale (i.e. an absolute gain of 3% seems more meaningful than a gain of 0.03 if you don’t know the context of the reported decimal number). However due to the potential for confusion about absolute vs relative percentages, when we first report a percentage at lines 63-66 of the revision, we have added the following clarifying text to explain our use of percentages: “...the average expected gain in persistence (compared to doing nothing) from threat management was 0.033 per species (persistence is measured on a 0-1 scale, so this equates to an absolute gain of 3.3% per species; we express absolute changes in persistence using percentages for the remainder of the text)”.

Figures 1 – 5 also showed both the sum of values as well as per-species values, but then there is little discussion around why both were included, so I suggest to either simplify the figures and only show the sums, or add more discussion around why the per-species values were included.

Response: Accepted. The reason we reported both summed values as well as per-species values was to ensure that the results were not biased only by the number of species affected by a KTP. For example, in figure 1, the summed gain in persistence for high frequency fire was large (4th highest KTP) due to the very large number of impacted species (affects 972 species). However at the per-species level, the benefit of managing fire under current uncertainty was negligible (see text lines 68-71). By reporting the per-species as well as the summed species results, we can see where we can get the highest overall persistence gains as well as where we can get high per-species gains. This highlights that objectives matter and also allows decision makers with either objective to use the results accordingly. We have added more discussion about why we included both objectives into the methods section, lines 378-382 of the revision)

Is figure 3 showing some of the information also shown in figure 1? This was not quite clear to me.

Response: Clarification: The former figure 3 (now figure 4) differs from figure 1 because of the level at which it is computed. Figure 1 is an aggregated measure which is averaged over all the species groups affected by the KTP. The former figure 3 shows the benefits of managing the most beneficial of these component species groups (the benefits of managing all species groups are reported in the Supporting Information S2). This is explained at the start of the section containing the former figure 3 (lines 122-126 of the revision). This should also be clearer due to the addition of our new figure 3, which illustrates the VOI analysis results for a single KTP and clearly shows the component species groups and the aggregate KTP level results.

Some abbreviations or terms were not explained in the main text or in the supplementary material – what does TEC stand for? Please add what this means to the figure captions too. What are critical weight mammals? Could KTP be changed to threats?

Response: Accepted. ‘TEC’ stands for ‘threatened ecological community’; we have replaced the acronym with ‘ecological community’ throughout the text and added the definition to the captions of figures 4 and 5 in the revision.

Critical weight range mammals are mammals with body mass 35g-5500g; these species are undergoing rapid extinctions within Australia, largely attributable to the impacts of invasive

mesopredators (cats and foxes). We have added a citation to a paper which defines and explains the critical weight range mammals (Burbidge & McKenzie 1989) (lines 71-72 of the revision).

KTP has been changed to threat throughout the main text, with the exception of lines 45-55, where KTPs are first defined using the New South Wales context—here it is necessary to use ‘KTP’ rather than ‘threat’ so that it is clear what we mean when we refer to ‘threats’ later in the manuscript. We have kept ‘KTP’ in the SI as it is more precise than the generic ‘threat’ which may help avoid confusion when we present the detailed methods and equations in the SI.

I found the section on EVPXI difficult to follow because of how the information is presented in figure 5. It would help if figure 5 would show the 3 bars of the uncertainties next to each other. Where is the green bar shown in the legend?

Response: Accepted. We have changed the figure so that the bars are adjacent to each other. The green bar does not show because the uncertainty about the threat removal is negligibly small. This was mentioned in the figure caption, but we have now elaborated the caption to ensure this is clearer to the reader: “In this study, reducing uncertainty about threat removal (green in the figure legend) has negligible value of information so does not appear in the plot.” (lines 231-232 of the revision)

Fire seems to stand out from the other threats in that there is high uncertainty around how it can be managed. However, it seems that this is based on the expert judgement by one person only. This would be worth mentioning in the discussion.

Response: Accepted. We have added the following paragraph to the discussion (lines 312-321): “A limitation of elicitation in threatened species management is that sometimes few experts are available. For example, here the expected impacts of high frequency fire are obtained from one expert (see Table 3 in Supporting Information S1). In the absence of additional information, it is better to use the expert information available and critically examine the results than to ignore the threat. For example, here the findings of high value of information for the fire threat are likely to be robust, as: (i) most of the potential value of fire management comes from the sheer number of species affected by fire rather than from expert-predicted benefits of management (the predicted gains from management were very low for most species groups affected by fire), and (ii) the uncertainty which creates the value of information is well documented for fire (Bradstock 2008; Driscoll *et al.* 2010).

This is a well-written manuscript, particularly the introduction and section 3.6 are to the point and cover what is most important, with appropriate use of references. The authors make a good case for why this is important work. The abstract is clear and concise.

References cited in this response:

Bottrill, M.C., Joseph, L.N., Carwardine, J., Bode, M., Cook, C., Game, E.T., Grantham, H., Kark, S., Linke, S., McDonald-Madden, E., Pressey, R.L., Walker, S., Wilson, K.A. & Possingham, H.P. (2008) Is conservation triage just smart decision making? *Trends in Ecology & Evolution*, **23**, 649-654.

- Bradstock, R.A. (2008) Effects of large fires on biodiversity in south-eastern Australia: disaster or template for diversity? *International Journal of Wildland Fire*, **17**, 809-822.
- Burbidge, A.A. & McKenzie, N.L. (1989) Patterns in the modern decline of western Australia's vertebrate fauna: Causes and conservation implications. *Biological Conservation*, **50**, 143-198.
- Cahill, D.M., Rookes, J.E., Wilson, B.A., Gibson, L. & McDougall, K.L. (2008) Phytophthora cinnamomi and Australia's biodiversity: impacts, predictions and progress towards control. *Australian Journal of Botany*, **56**, 279-310.
- Carwardine, J., Martin, T.G., Firn, J., Reyes, R.P., Nicol, S., Reeson, A., Grantham, H.S., Stratford, D., Kehoe, L. & Chadès, I. (2019) Priority Threat Management for biodiversity conservation: A handbook. *Journal of Applied Ecology*, **56**, 481-490.
- Doherty, T.S. & Ritchie, E.G. (2017) Stop Jumping the Gun: A Call for Evidence-Based Invasive Predator Management. *Conservation Letters*, **10**, 15-22.
- Driscoll, D.A., Lindenmayer, D.B., Bennett, A.F., Bode, M., Bradstock, R.A., Cary, G.J., Clarke, M.F., Dexter, N., Fensham, R., Friend, G., Gill, M., James, S., Kay, G., Keith, D.A., MacGregor, C., Russell-Smith, J., Salt, D., Watson, J.E.M., Williams, R.J. & York, A. (2010) Fire management for biodiversity conservation: Key research questions and our capacity to answer them. *Biological Conservation*, **143**, 1928-1939.
- Lazenby, B.T., Mooney, N.J. & Dickman, C.R. (2015) Effects of low-level culling of feral cats in open populations: a case study from the forests of southern Tasmania. *Wildlife Research*, **41**, 407-420.
- Shih, T.-H. & Fan, X. (2009) Comparing response rates in e-mail and paper surveys: A meta-analysis. *Educational Research Review*, **4**, 26-40.
- Speirs-Bridge, A., Fidler, F., McBride, M., Flander, L., Cumming, G. & Burgman, M. (2010) Reducing Overconfidence in the Interval Judgments of Experts. *Risk Analysis*, **30**, 512-523.
- Sutherland, W.J., Pullin, A.S., Dolman, P.M. & Knight, T.M. (2004) The need for evidence-based conservation. *Trends in Ecology & Evolution*, **19**, 305-308.
- Walsh, J.C., Wilson, K.A., Benshemesh, J. & Possingham, H.P. (2012) Unexpected outcomes of invasive predator control: the importance of evaluating conservation management actions. *Animal Conservation*, **15**, 319-328.

Reviewers' Comments:

Reviewer #1:

Remarks to the Author:

The authors have provided a detailed and thoughtful response to the reviews, including fairly extensive discussion supported by graphs of some of the issues raised.

I am very satisfied with the response and revision and have no remaining suggestions for improvement.

Michael C. Runge, USGS Patuxent Wildlife Research Center

Reviewer #3:

Remarks to the Author:

The authors addressed all concerns that I had previously. I can see now why they explained the decision problem before the data analysis section. I also like the new figure 3 (you could increase the font size of the y-axis for the EV uncertainty and certainty, but that is all), and other revisions/explanations to figures. Your explanation of why a two threat comparison is difficult is also good, and your addition about the limitation of having just one expert on fire as a threat. Overall this is an exciting manuscript which will be very useful in advancing the use of VoI for conservation decision-making, and how we go about threatened species management in the future.

REVIEWERS' COMMENTS:

Reviewer #1 (Remarks to the Author): The authors have provided a detailed and thoughtful response to the reviews, including fairly extensive discussion supported by graphs of some of the issues raised.

I am very satisfied with the response and revision and have no remaining suggestions for improvement.

Michael C. Runge, USGS Patuxent Wildlife Research Center

Reviewer #3 (Remarks to the Author):

The authors addressed all concerns that I had previously. I can see now why they explained the decision problem before the data analysis section. I also like the new figure 3 (you could increase the font size of the y-axis for the EV uncertainty and certainty, but that is all), and other revisions/explanations to figures. Your explanation of why a two threat comparison is difficult is also good, and your addition about the limitation of having just one expert on fire as a threat.

Overall this is an exciting manuscript which will be very useful in advancing the use of Vol for conservation decision-making, and how we go about threatened species management in the future.

Author response to reviewers: We thank all the reviewers for their positive review of our manuscript.